# The airway microbiota of neonates colonized with asthma-associated pathogenic bacteria

Jonathan Thorsen [1], Xuan Ji Li [2], Shuang Peng[2], Rikke Bjersand Sunde[1,3], Shiraz A. Shah [1], Madhumita Bhattacharyya[4], Casper Sahl Poulsen[1], Christina Egeø Poulsen[1], Cristina Leal Rodriguez [1], Michael Widdowson [1], Avidan Uriel Neumann[4,5], Urvish Trivedi[2], Bo Chawes [1], Klaus Bønnelykke [1], Hans Bisgaard[1,8], Søren J. Sørensen [2,7] ✉ & Jakob Stokholm [1,3,6,7] ✉

Culture techniques have associated colonization with pathogenic bacteria in the airways of neonates with later risk of childhood asthma, whereas more recent studies utilizing sequencing techniques have shown the same phenomenon with specific anaerobic taxa. Here, we analyze nasopharyngeal swabs from 1 month neonates in the COPSAC$_{2000}$ prospective birth cohort by 16S rRNA gene sequencing of the V3-V4 region in relation to asthma risk throughout childhood. Results are compared with previous culture results from hypopharyngeal aspirates from the same cohort and with hypopharyngeal sequencing data from the later COPSAC$_{2010}$ cohort. Nasopharyngeal relative abundance values of *Streptococcus pneumoniae*, *Haemophilus influenzae*, and *Moraxella catarrhalis* are associated with the same species in the hypopharyngeal cultures. A combined pathogen score of these bacteria's abundance values is associated with persistent wheeze/asthma by age 7. No other taxa are associated. Compared to the hypopharyngeal aspirates from the COPSAC$_{2010}$ cohort, the anaerobes *Veillonella* and *Prevotella*, which have previously been implicated in asthma development, are less commonly detected in the COPSAC$_{2000}$ nasopharyngeal samples, but correlate with the pathogen score, hinting at latent community structures that bridge current and previous results. These findings have implications for future asthma prevention efforts.

Asthma is one of the most common chronic diseases in childhood and is influenced by risk factors in pregnancy and early infancy[1]. Many of these are related to microbial exposures[2–4] and prospective cohort studies have shown associations between the early-life airway and gut microbiota and asthma development[5–11]. These findings have garnered immense interest since they may increase our understanding of asthma disease biology and lead to novel opportunities for asthma prevention[12].

In particular, a study from the Copenhagen Prospective Studies on Asthma in Childhood 2000 (COPSAC$_{2000}$) cohort initiated the interest in the early airway microbiota as a predictor of disease. That study demonstrated an association between colonization with the

---

[1]COPSAC, Copenhagen Prospective Studies on Asthma in Childhood, Herlev and Gentofte Hospital, University of Copenhagen, Copenhagen, Denmark. [2]Department of Biology, Faculty of Science, University of Copenhagen, Copenhagen, Denmark. [3]Department of Pediatrics, Slagelse Hospital, Slagelse, Denmark. [4]Chair of Environmental Medicine, Faculty of Medicine, University of Augsburg, Augsburg, Germany. [5]Institute of Environmental Medicine, Helmholtz Munich, Munich, Germany. [6]Department of Food Science, Faculty of Science, University of Copenhagen, Frederiksberg C, Denmark. [7]These authors jointly supervised this work: Søren J. Sørensen, Jakob Stokholm. [8]Deceased: Hans Bisgaard. ✉e-mail: sjs@bio.ku.dk; stokholm@copsac.com

pathogenic bacteria *Streptococcus pneumoniae, Haemophilus influenzae,* and *Moraxella catarrhalis* in the hypopharynx of 1-month-old neonates, detected by cultures, and the development of asthma and asthma exacerbations by age 5[5]. Later, in the subsequent COPSAC[2000] cohort, we found similar associations between the 1-month hypopharyngeal microbiota, especially the anaerobic taxa *Veillonella* and *Prevotella*, and the risk of asthma by age 6[7]. In that study, we applied next-generation sequencing of 16S ribosomal RNA gene amplicons—in short, 16S sequencing. However, it has remained unclear whether the differences in asthma-associated bacteria between these studies were due to differences in sampling, the applied detection technique (culturing vs sequencing), or simply due to inherent cohort differences. Notably, while the nasopharyngeal and hypopharyngeal communities are in constant exchange of microbes, these niches are also preferentially colonized by particular species[13]. Since the first COPSAC[2000] study, the prevalence and availability of culture-independent techniques, including 16S sequencing, has increased substantially, allowing the detection of many more species, including those that are not routinely found by culturing.

Here, we present 16S sequencing data from the COPSAC[2000] cohort, obtained by processing of biobank stored nasopharyngeal swabs. Our aim was to use this data to examine whether the pathogen culture association with asthma could be re-established using sequencing data from these anatomically distinct but adjacently collected samples. Further, we examine whether any other taxa associated with asthma could be identified using the more sensitive sequencing method—in particular a possible replication of the COPSAC[2010] *Veillonella* and *Prevotella* associations.

An overview of the different cohorts, sampling locations, and techniques applied is shown in Fig. 1.

## Results

### Nasopharyngeal microbiota

We successfully sequenced 285 1-month nasopharyngeal swab samples from the children of the COPSAC[2000] cohort to a depth of at least 2000 reads, removing four samples due to low read depth. The median sequencing depth was 60,723, interquartile range (IQR) [45,384; 79,958]. After quality control, 4023 ASVs were retained and the samples had a median richness of 36 ASVs, IQR [23; 48]. The children with samples available were similar to children with no samples available with respect to their baseline characteristics, only differing significantly by season of birth, lower rates of maternal smoking, and marginally by more often having siblings (Table 1).

### Nasopharyngeal microbiota, hypopharyngeal cultures, and asthma

First, we compared the sequenced nasopharyngeal swabs to the hypopharyngeal aspirate culture results originally reported to be associated with asthma by age 5[5], in order to ensure that there was agreement between these two sample sets and methods before comparing their associations with asthma. In total, 244 children had both sequencing and culture data available. We compared the results from each species identified in the cultures (i.e., *Streptococcus pneumoniae, Haemophilus influenzae, Moraxella catarrhalis,* and *Staphylococcus aureus*) and found that the relative abundance in the sequencing data was strongly linked to the culture results (*S. pneumoniae* colonized vs.

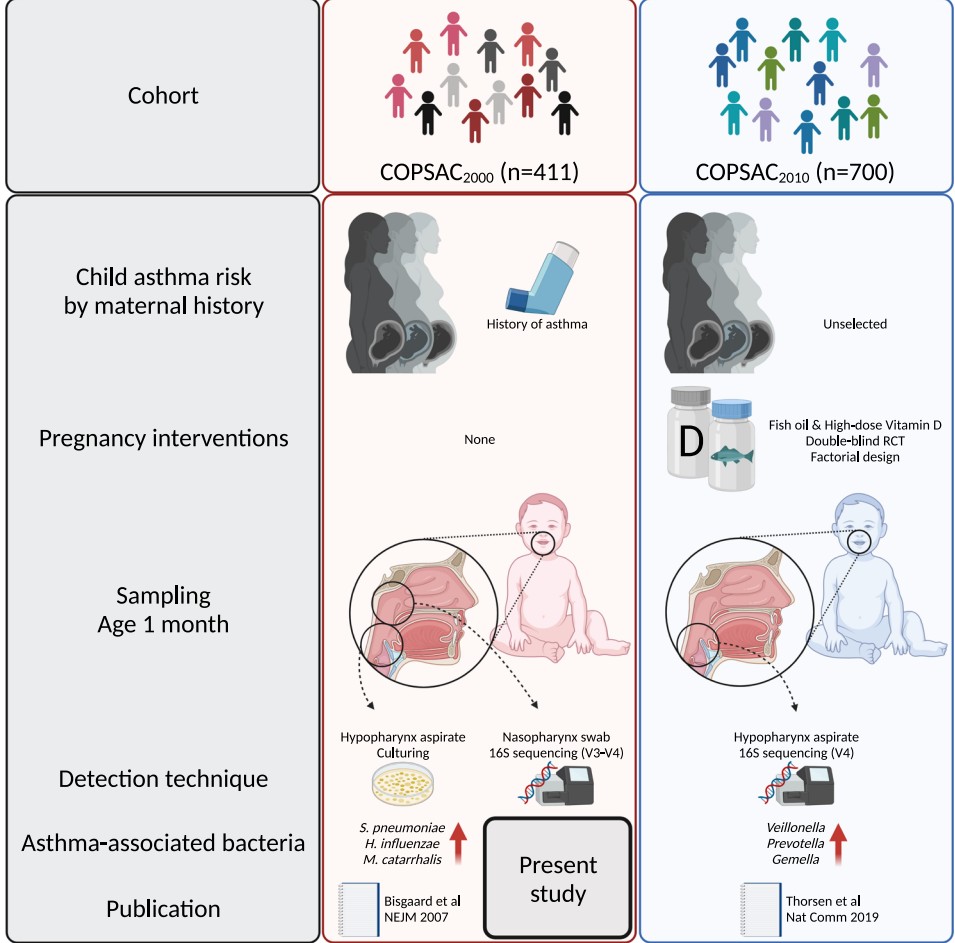

**Fig. 1 | Study overview.** Characteristics and previously published results on airway bacteria and asthma from the two COPSAC cohorts[5,7]. Created with BioRender.com.

**Table 1 | Baseline characteristics of the children of the COPSAC$_{2000}$ cohort, comparing the children who had nasopharyngeal swab samples successfully obtained and sequenced with those who did not**

| Characteristic | N | Sample available, N = 285[a] | No sample available, N = 126[a] | p value[b] |
|---|---|---|---|---|
| Sex | 411 | | | 0.5 |
| Female | | 141 (49%) | 67 (53%) | |
| Male | | 144 (51%) | 59 (47%) | |
| Ethnicity | 411 | | | 0.8 |
| Northern European | | 276 (97%) | 121 (96%) | |
| Other | | 9 (3.2%) | 5 (4.0%) | |
| Delivery mode | 411 | | | 0.4 |
| Cesarean section | | 63 (22%) | 23 (18%) | |
| Vaginal | | 222 (78%) | 103 (82%) | |
| Gestational age (weeks) | 411 | 40.43 (39.29, 41.43) | 40.29 (39.43, 41.71) | 0.5 |
| Birth weight (kg) | 411 | 3.50 (3.20, 3.82) | 3.55 (3.20, 3.80) | 0.8 |
| Birth length (cm) | 411 | 52.00 (51.00, 54.00) | 52.00 (51.00, 54.00) | 0.9 |
| Season of birth | 411 | | | 0.00006 |
| Autumn | | 64 (22%) | 55 (44%) | |
| Spring | | 70 (25%) | 16 (13%) | |
| Summer | | 85 (30%) | 26 (21%) | |
| Winter | | 66 (23%) | 29 (23%) | |
| Living environment | 411 | | | 0.9 |
| Rural | | 147 (52%) | 66 (52%) | |
| Urban | | 138 (48%) | 60 (48%) | |
| Older siblings, any | 386 | 113 (42%) | 38 (32%) | 0.053 |
| Missing data | | 18 | 7 | |
| Furred pet at home | 389 | 78 (29%) | 37 (31%) | 0.6 |
| Missing data | | 14 | 8 | |
| Maternal smoking at 1 month | 354 | 39 (16%) | 27 (26%) | 0.027 |
| Missing data | | 36 | 21 | |
| Breastfeeding at 1 month | 367 | 210 (82%) | 93 (85%) | 0.5 |
| Missing data | | 28 | 16 | |
| Social circumstances PCA score | 382 | −0.07 (−0.70, 0.60) | −0.15 (−0.81, 0.51) | 0.3 |
| Missing data | | 20 | 9 | |
| Antibiotics during pregnancy | 411 | 83 (29%) | 42 (33%) | 0.4 |

[a]n (%); median (interquartile range).

[b]Pearson's Chi-squared; Fisher's exact test; Wilcoxon rank-sum test. P values are two-sided and not adjusted for multiple comparisons.

not colonized, median [IQR], 7.18% [0.53%; 21.1%] vs 0.03% [0.00%; 0.62%], *H. influenzae* 0.00% [0.00%; 42.4%] vs 0.00% [0.00%; 0.00%], *M. catarrhalis* 3.05% [0.42%; 11.1%] vs 0.00% [0.00%; 0.00], *S. aureus* 44.2% [0.45%; 79.0%] vs 0.15% [0.08%; 0.32%], Wilcoxon tests, all $p < 0.0001$, see Fig. 2a and Supplementary Fig. 1). However, a few samples with positive hypopharynx cultures did not reveal the same species in the nasopharyngeal sequencing data. Many samples that were culture-negative for a specific species were positive in the sequencing data. We found a high degree of species-level specificity when comparing the culture results with the most common ASVs from the same genus, for all four species (Supplementary Fig. 2). Having established that these two samples were to an extent comparable, we

progressed to our main aim of investigating if the association between neonatal pathogen colonization with *S. pneumoniae, H. influenzae*, and *M. catarrhalis* and asthma[5] could be recapitulated with the sequencing data by summing the relative abundance values of these three species into a combined pathogen score and comparing against persistent wheeze/asthma (Fig. 2b). We found the pathogen score to be associated with an increased risk of later asthma (Cox regression, Hazard Ratio per standard deviation (HR) 1.50, 95% Confidence Interval (CI) [1.12; 2.01], $p = 0.0066$, $n = 285$). This association was similar between both sexes (male, HR 1.52 [1.07; 2.16], $p = 0.02$, $n = 144$; female, HR 1.58 [0.94; 2.65], $p = 0.08$, $n = 141$; interaction $p = 0.90$). The pathogen score was associated with delivery mode and having siblings at home (Supplementary Table 1). However, adjusting for these and other potential important covariates did not change the association with asthma (adjusted Hazard Ratio (aHR) 1.48 [1.06; 2.05], $p = 0.02$, $n = 241$ − adjusted for siblings, delivery mode, living environment, maternal smoking, season of birth, sex, maternal antibiotic use during pregnancy, breastfeeding at 1 month, gestational age at birth). The association disappeared when repeating the analysis only among the children with no pathogenic bacteria detected by cultures (HR 0.93 [0.60; 1.46], $p = 0.44$, $n = 188$) or in all children when adjusting for the presence of pathogenic bacteria (HR 1.06 [0.73; 1.52], $p = 0.76$), indicating that the asthma association was adequately captured by the culture results. The culturing-asthma association remained significant when adjusting for the pathogen score (aHR 3.50 [1.69; 7.25], $p = 0.0007$, $n = 244$), which seemingly indicated that the culture result was a stronger predictor of the clinical endpoint than the sequencing. Finally, the pathogen score was also associated with other asthma-related endpoints, including the less strict recurrent wheeze phenotype as well as early exacerbations (Fig. 2c). Of note, while the pathogen colonization was associated with hospitalization for wheeze (by age 5, HR 3.88 [1.92; 7.86], $p = 0.00016$, $n = 319$), the pathogen score only showed a non-significant trend.

**Nasopharyngeal diversity, differential abundance, and asthma**
Having established a good correspondence between the hypopharyngeal cultures and the microbiota in nasopharyngeal aspirates, and that both were associated with asthma, we next investigated whether any other parts of the nasopharyngeal microbiome were also associated with asthma development. We quantified the diversity of the microbiome using the Shannon diversity index and Faith's phylogenetic diversity, neither of which was associated with asthma (Cox regression, Shannon: HR 0.72 [0.42; 1.22], $p = 0.22$; Faith: HR 1.02 [0.88; 1.18], $p = 0.78$; $n = 285$). We compared the overall microbiome composition of children who developed asthma by age 7 with those who did not, which were similar (PERMANOVA, unweighted UniFrac, $F = 1.21$, $p = 0.17$; log weighted UniFrac, $F = 1.17$, p = 0.25; $n = 221$). We selected the 35 most common taxa (see Supplementary Fig. 3) for differential abundance testing using two complementary methods (DESeq2 and Cox regression), see Fig. 3. Among the nominally significant taxa, only *Haemophilus influenzae, Streptococcus pneumoniae, Moraxella catarrhalis* and *Moraxella lincolnii* were identified; of these, only the first two were significant in either test after false discovery rate (FDR) correction. Thus, apart from the species identified in cultures, only *Moraxella lincolnii* came up as a potential novel species associated with asthma. *M. catarrhalis* and *M. lincolnii* are two distinct species, and to confirm that their 16S rRNA gene sequences are indeed different enough to confidently separate them in our data, we investigated their genetic similarity. We found them distinctly identifiable in the V3-V4 region used in this study based on 16S rRNA genes in reference genomes (Supplementary Fig. 4).

**Nasopharyngeal *Veillonella* and *Prevotella* colonization**
In the COPSAC$_{2010}$ study, we found that *Veillonella, Prevotella,* and some accompanying taxa in hypopharyngeal aspirates at age 1 month

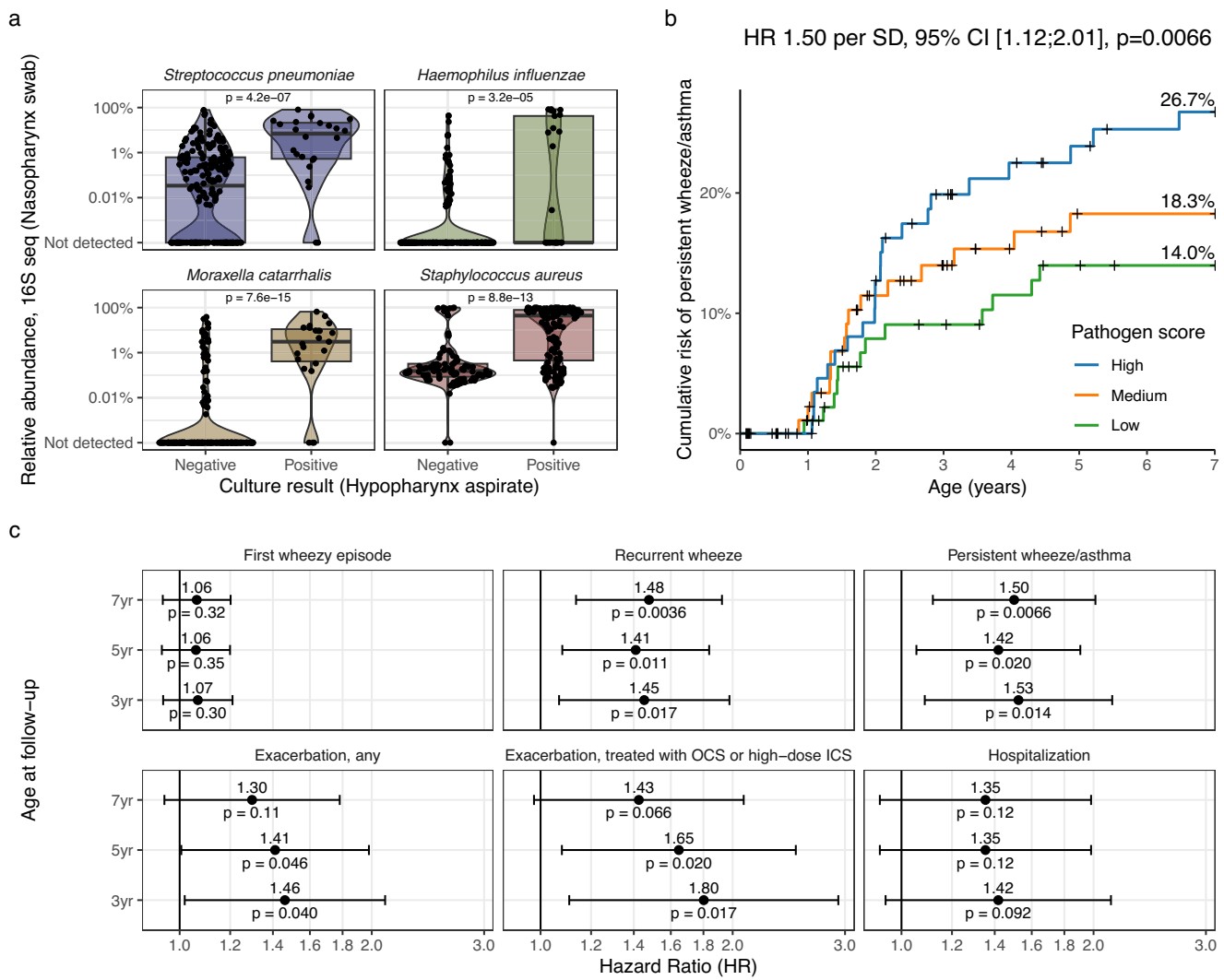

**Fig. 2 | Nasopharyngeal microbiota associates with hypopharyngeal culture and childhood asthma. a** Comparison of results from culturing of hypopharyngeal aspirates with relative abundance values from 16S rRNA gene sequencing of nasopharyngeal swabs, for each of the four species identified in the cultures. Box plots indicate median (middle line), 25th, 75th percentile (box), and 1.5× inter-quartile range (whiskers). *P* values are from Wilcoxon tests. **b** Kaplan–Meier curve showing the risk of asthma by each tertile of summed relative abundance of *Streptococcus pneumoniae*, *Haemophilus influenzae*, and *Moraxella catarrhalis*

(pathogen score) with 7-year risk estimates per curve. Censoring is marked with +. **c** Associations between the pathogen score and asthma-related endpoints at 3, 5, and 7 years. All Hazard Ratios (HR, dots) are quantified per standard deviation (SD) increase in the pathogen score with corresponding 95% confidence intervals (error bars), calculated with Cox proportional hazards regression. OCS oral corticosteroid, ICS Inhaled corticosteroid. $N = 285$. Data from the COPSAC$_{2000}$ cohort. *P* values are two-sided and not adjusted for multiple comparisons.

were associated with asthma development by age 6 years. However, in the nasopharyngeal swab samples in the present study, these taxa were much less commonly identified, see Fig. 4a, b. *Veillonella* was identified in 40.4% of the COPSAC$_{2000}$ samples vs 77.2% of the COPSAC$_{2010}$ samples, and *Prevotella* in 17.9% vs 42.3%. In COPSAC$_{2000}$, at least one of these two genera was detected in 47% of the samples (134/285). At the species level, *V. montpellierensis*, *V. dispar,* and *V. atypica* were found in more than 10% of the samples, whereas the most common *Prevotella* species was only found in 6.67% of the samples (Supplementary Table 2). Detection of either of these two genera was not associated with persistent wheeze/asthma by age 7 years, nor were they differentially abundant at the genus level (Fig. 4c–e). Adjusting analyses for the same covariates mentioned above did not change the results, but mutually adjusting the presence/absence of *Veillonella/ Prevotella* and the pathogen score resulted in attenuation of the estimate for *Veillonella/Prevotella* but not the pathogen score (Fig. 4e). None of the individual species were significant in the overall

differential abundance analysis described above (Fig. 3), although only *V. montpellierensis* and *V. dispar* were sufficiently common to be included in the test (Supplementary Fig. 3).

Next, we compared their relative abundance with the pathogen score, which showed that both genera were significantly associated (*Veillonella*, Spearman Rho 0.25, $p < 0.0001$; *Prevotella*, Spearman Rho 0.14, $p = 0.016$, $n = 285$) and the same was found for the individual species *P. nanceiensis, V. montpellierensis, V. parvula, V. atypica* and *V. dispar*, but not *P. melaninogenica* (Supplementary Fig. 5). Lastly, we sought to clarify whether this apparent relationship was stronger than for other taxa. We found that *Veillonella* and *Prevotella* were included in an apparent cluster with *Streptococcus* and *Haemophilus* (Fig. 5 and Supplemental Fig. 6). Further, *Veillonella,* and *Prevotella* were among the strongest correlated with the pathogen score apart from *Streptococcus*, *Haemophilus*, and *Moraxella*, and especially *Prevotella* had an even stronger correlation with the pathogen score in the children for which it was positive (Supplemental Fig. 6).

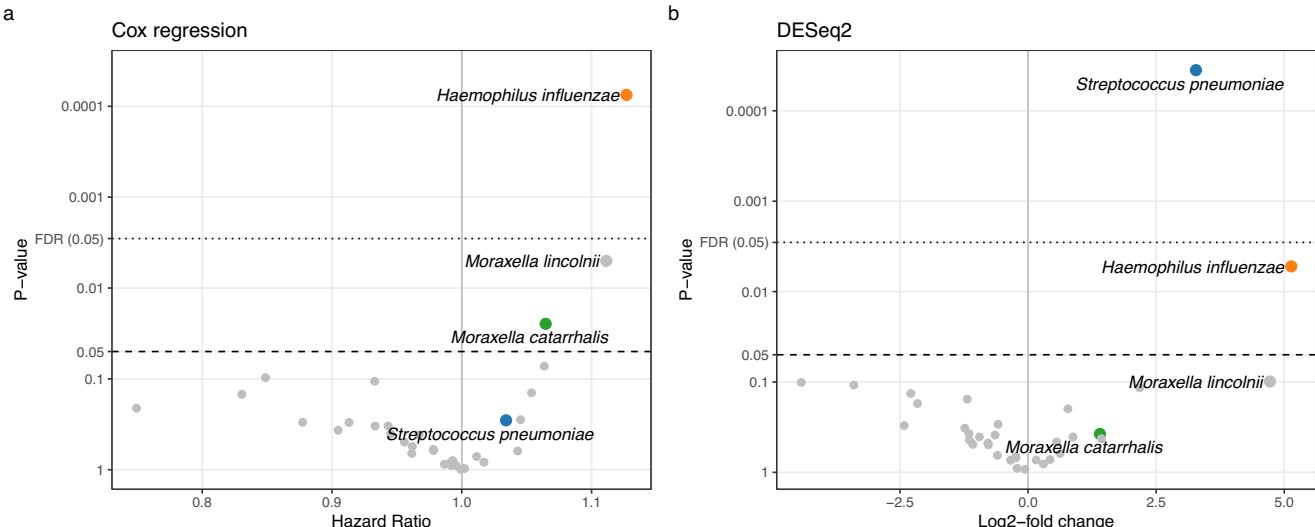

**Fig. 3 | Individual taxa and childhood asthma risk.** Differential abundance tests of individual bacterial species and asthma by age 7, using two different statistical models; **a** Cox regression of time-to-asthma quantified as Hazard Ratio per log2 increase in species relative abundance and **b** DESeq2 analysis comparing abundance of the species between children who later developed asthma vs those who did not. The same set of four species, that are significant in either analysis, are highlighted. FDR, false discovery rate. $N = 285$. Data from the COPSAC$_{2000}$ cohort. $P$ values are two-sided and not adjusted for multiple comparisons—instead, the False Discovery Rate (FDR) cutoff is shown with a dotted line.

## Discussion

We found that the nasopharyngeal microbiota, assessed using 16S sequencing, was correlated with culturing of pathogenic airway bacteria from hypopharyngeal aspirates in the high-risk COPSAC$_{2000}$ cohort of children born to mothers with a history of asthma. A pathogen score, based on nasopharyngeal relative abundance of *Streptococcus pneumoniae*, *Haemophilus influenzae,* and *Moraxella catarrhalis*, was associated with persistent wheeze/asthma by age 7 and exacerbations by age 5, in line with the previously published association with hypopharyngeal colonization. Stratifying or adjusting for colonization left no additional association for the pathogen score. *Moraxella lincolnii* was the only other species that was significantly associated with asthma, but it did not remain so after FDR correction. Compared to the hypopharyngeal aspirates from the COPSAC$_{2010}$ cohort, *Veillonella* and *Prevotella* species were less commonly detected in the data and not associated with asthma but were positively associated with the pathogen score.

The main strength of the study is the systematically and prospectively assessed endpoints of wheeze, asthma, and exacerbations diagnosed solely by study physicians through detailed symptom recording and frequent clinical visits. This approach ensures consistency of the phenotype and avoids any differences between diagnoses set by community doctors. Furthermore, it is a major strength that the study uses early-life samples taken before the onset of the disease, combined with a well-resolved longitudinal clinical follow-up. The well-described cohort allowed us to adjust our findings for many potential confounders, which did not substantially change the results.

This study used next-generation sequencing, which is able to detect many more bacteria than the culturing applied in the previous study[5], and can capture bacteria that are not alive or cultivable. We acknowledge the usual limitations to 16S sequencing: Copy number variation, amplification bias, and potential read errors and contamination. The 16S rRNA gene is only found in bacteria and archaea, which means that viruses and fungi are not detected although they may also be important players in the early-life airway microbiome and influence the developing immune system[14,15]. Comparing the two COPSAC cohorts' sequencing data was further limited by using different primer sets and the amplification of different regions of the 16S rRNA gene (V4 vs V3-V4). Species annotations of the ASVs should be

considered putative since they are based on an algorithm (AnnotIEM) combining multiple database sources. However, combining our sequencing data with the mock community in positive controls and the hypopharyngeal aspirate cultures showed good sensitivity and specificity for the four cultured species, even though they were sampled from different parts of the upper respiratory tract. We did not have any original sample material left from the COPSAC$_{2000}$ hypopharynx aspirates, which would have made an even more precise comparison with the previous asthma-associating studies[5,7] possible (Fig. 1). However, data from the nasopharyngeal swabs correlated with the cultured pathogenic species, highlighting the continuity of colonization in the respiratory tract[16–19] in addition to local adaptation[13].

The nasopharyngeal swab samples have been stored for ~20 years in a biobank before processing. Despite this long storage time and other major differences between sample types, we found good correspondence with the hypopharyngeal cultures, which were processed immediately after sampling, indicating that the storage time did not have major detrimental effects on the quality of the samples. However, there may be taxa that have been selectively degraded in the samples, which would skew the overall composition compared to the in vivo community. Similarly, the samples underwent culture for mycobacteria before freezing to −80 °C, which could contribute to the degradation of anaerobic bacteria like *Veillonella* and *Prevotella*. Notably, such skewness, like other technical biases, would equally affect all samples and thus not interfere with associations with clinical phenotypes. However, they could interfere with the comparisons made with other cohorts, including COPSAC$_{2010}$. Importantly, our study was not designed to compare different upper airway sampling locations or detection methods, which have been performed elsewhere[20–22], but rather to examine early-life risk factors for disease, in particular asthma.

The most important result of the study is that the association between pathogenic bacteria and asthma remained the same as in the previous study[5] despite applying a different sampling technique (swabs vs aspirates), sampling site (nasopharynx vs hypopharynx), and detection technique (16S sequencing vs culturing), which underscores the robustness of the association in this cohort. When mutually adjusting the highly correlated culture and sequencing variables, or stratifying for the culture results, the culture variable seemed to carry

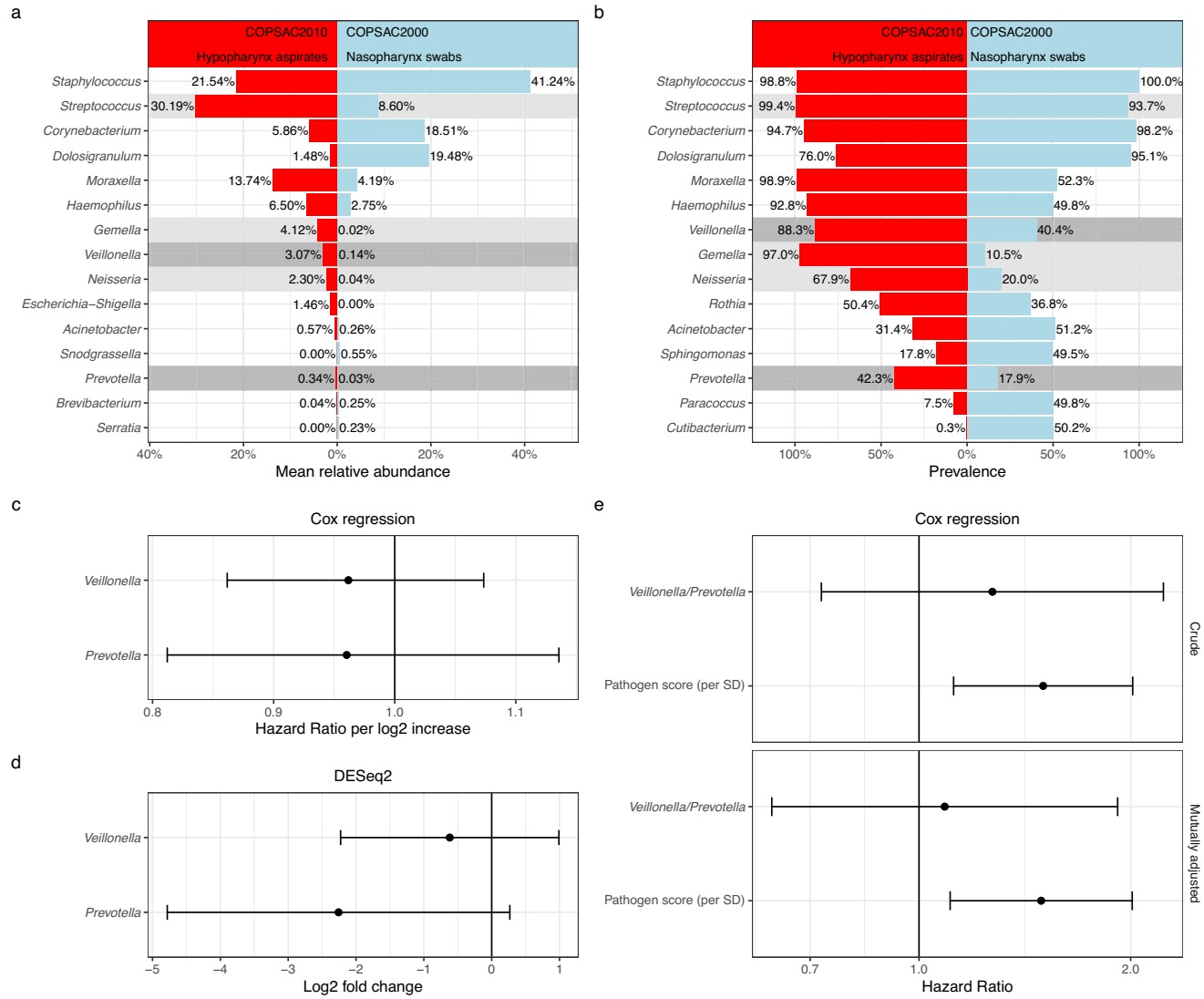

**Fig. 4 | Apparent observed cohort differences in the detection of *Veillonella* and *Prevotella*. a, b** Comparison of COPSAC$_{2010}$ hypopharyngeal aspirates and COPSAC$_{2000}$ nasopharyngeal swabs by genus names. *Veillonella* and *Prevotella* are highlighted due to their significant association with asthma in COPSAC$_{2010}$ (dark gray). *Gemella, Streptococcus,* and *Neisseria* also contributed to the bacterial asthma score, but were not individually significant after FDR correction (light gray). Different regions were amplified in the two cohorts (COPSAC$_{2000}$−V3-V4; COPSAC$_{2010}$−V4), but the bioinformatics and data processing pipeline was identical (DADA2 + AnnotIEM, see methods). **c, d** Genus abundance values of *Veillonella* and *Prevotella* are not associated with persistent wheeze/asthma by age 7 in COPSAC$_{2000}$. **e** *Veillonella* and/or *Prevotella* presence is not associated with persistent wheeze/asthma by age 7 in COPSAC$_{2000}$. Their association estimate attenuated after mutual adjustment with the pathogen score, which in contrast did not attenuate. Dots indicate estimates, error bars indicate 95% confidence intervals (CI). N = 285 (COPSAC$_{2000}$), N = 641 (COPSAC$_{2010}$). P values are two-sided and not adjusted for multiple comparisons.

the strongest part of the joint association. We speculate that this could be related to the absolute abundance of these bacteria, which was not assessed in the relative abundance space of the sequencing data.

As a potentially novel finding, *Moraxella lincolnii*[23] was positively associated with asthma in our study, although not passing FDR correction. No previous studies have associated this species with asthma, but it has been found to be inversely associated with respiratory infections[24] and otitis media[25] in children, and associated with a better outcome in critically ill adults with COVID-19[26]. Furthermore, longitudinal studies have shown that *M. lincolnii* is commonly found in healthy children later in childhood[27,28]. Taken together, these data indicate a more commensal role of this species.

The two COPSAC$_{2010}$ asthma-associated anaerobic genera *Veillonella* and *Prevotella*, were not replicated in the present study. The most prevalent *Veillonella* species were *V. montpellierensis, V. dispar,* and *V. atypica,* and for *Prevotella* were *P. melaninogenica, P. nanceiensis,* and

*P. veroralis*. However, both genera were considerably less frequent than in the COPSAC$_{2010}$ cohort. We speculate that a reason for this lower abundance could be the sampling location since these taxa are more commonly found in the oral cavity than the nasopharynx[21] and may therefore migrate more easily from the oral cavity to the hypopharynx, but this hypothesis cannot be tested directly in our data. Other potential reasons include the numerous differences in sample collection, handling, and storage−our study was not designed to infer the cause of such differences in observed abundance values, but rather proved an opportunity to attempt to reconcile the apparent differences in asthma associations between the cohorts. The low detection rate of these taxa in the nasopharynx would likely preclude any direct association with asthma from being found. However, we found that both genera and several of their species were significantly associated with the pathogen score from the relative abundance of *Streptococcus pneumoniae, Haemophilus influenzae,* and *Moraxella catarrhalis,* and

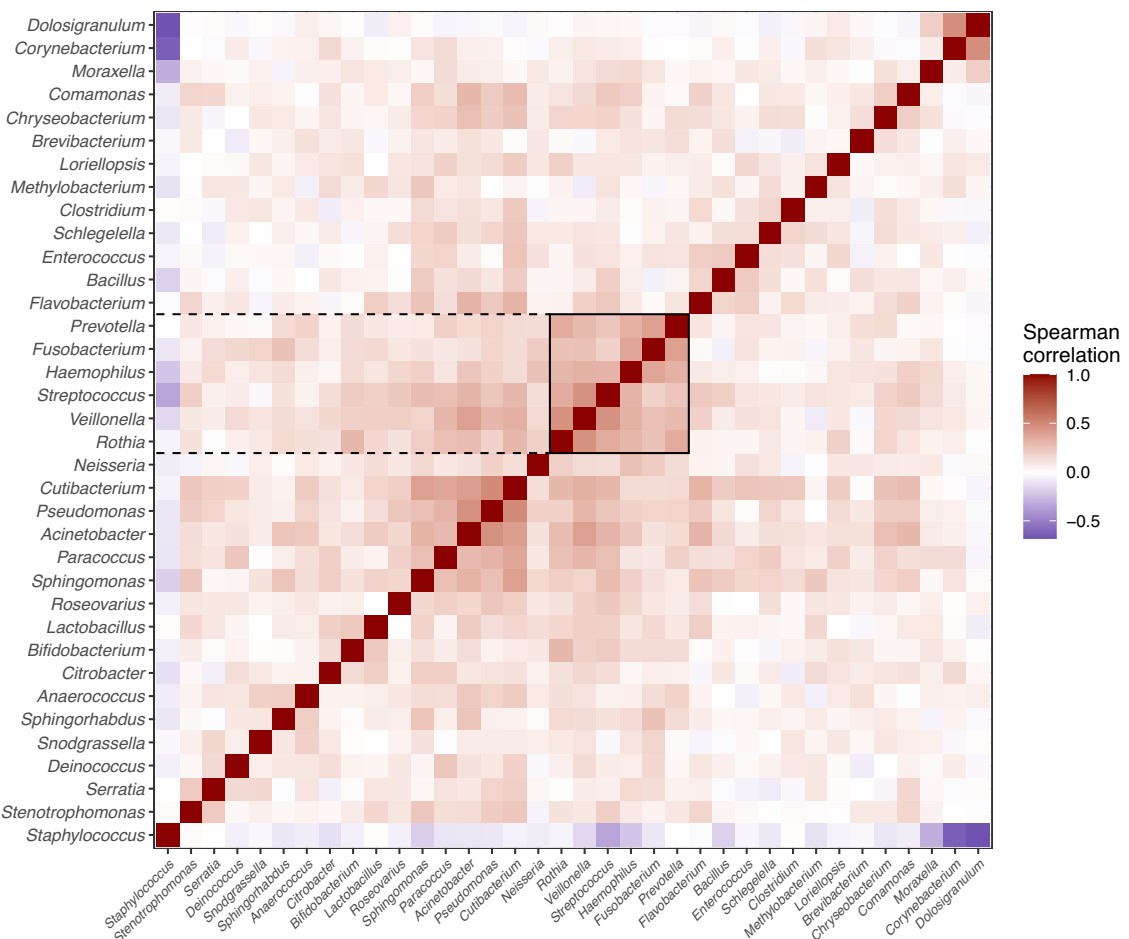

**Fig. 5 | Correlation structure between *Veillonella*, *Prevotella*, and pathogenic genera.** Correlation heatmap between the most common genera in COPSAC$_{2000}$, arranged by hierarchical clustering, using the Spearman rank correlation metric. We observe that *Veillonella* and *Prevotella* form a cluster with two of the three pathogen genera, *Streptococcus* and *Haemophilus*, which also includes a few other taxa. Highlighting was performed manually. See also Supplementary Fig. 6. $N = 285$.

that *Veillonella* and *Prevotella* were part of a correlation cluster with *Streptococcus* and *Haemophilus*. This suggests that some overlap in colonization and association with asthma may still be present between *Veillonella* & *Prevotella* and the three pathogenic bacteria of interest, which could be an expression of latent microbiota community structures or shared environmental determinants.

We can establish based on these results that the association between pathogenic bacteria and asthma in COPSAC$_{2000}$ recapitulated the culture-based findings independent of sampling location and method and detection method. While we did find overlaps between the taxa representing the pathogenic score and *Veillonella* and *Prevotella*, still some unresolved cohort differences exist. First, COPSAC$_{2000}$ is a high-risk cohort where all the mothers had a history of doctor-diagnosed asthma[29]. In contrast, the COPSAC$_{2010}$ cohort was unselected by design and had a rate of 30% self-reported asthma, which was higher than those who were invited but declined to participate[30]. Second, the COPSAC$_{2010}$ cohort was recruited with a wider geographical reach than COPSAC$_{2000}$[30,31], with more children living in rural settings, which could influence the microbiota composition and associations with asthma. Third, the rates of maternal smoking during pregnancy varied a lot between the cohorts, likely mirroring societal trends in Denmark[29,30]. Fourth, with regard to *S. pneumoniae*, Denmark introduced the pneumococcal vaccination as part of the childhood vaccination program in 2007[32]. While the infants in the COPSAC$_{2010}$ study were not yet vaccinated themselves at the time of sampling, the high child vaccination rates in the population (>95% in Denmark[33]) can influence the *S. pneumoniae* strains in the community[34] and potentially

affect associations with asthma. Similar changes may have occurred over time for other taxa. We adjusted our results for several of the above-mentioned factors, which did not change the conclusions, however, we cannot rule out residual confounding or unmeasured interactions between factors separating the two cohorts.

Despite these differences, there remains a solid association between the early-life airway microbiota and childhood wheeze/asthma, which has also been shown in other prospective birth cohorts than the two studied here. In the Childhood Asthma Study (CAS) cohort ($n = 234$, Perth, Australia, high risk for atopy), Teo and colleagues found an association between early nasopharyngeal *Streptococcus* colonization at 7–9 weeks and chronic wheeze at 5 years[6]. This was followed up with a study with extended longitudinal sample collection up to 5 years which among several findings revealed that frequent colonization during the first years of life with community subtypes associated with acute respiratory infections, dominated by *Moraxella*, *Streptococcus* or *Haemophilus*, was also associated with wheeze by age 5. A subanalysis clarified that in children with early allergic sensitization before age 2 years, these community types were associated with chronic wheeze, whereas in non-sensitized children, it was associated with a transient wheeze phenotype[35]. In the MARC-35 cohort of infants with severe bronchiolitis requiring hospitalization ($n = 842$, Boston, MA, US), a condition which is itself a strong risk factor for later wheeze and asthma, children colonized with high abundance of *Moraxella* or *Streptococcus* had increased risk of recurrent wheeze by age 3 years[36]. This was further elaborated in a follow-up study of a subset of these infants that showed an association of *S. pneumoniae*

with not only later asthma, but also airway metatranscriptomic enrichment of genes related to fatty acid and glycolysis pathways, which was also echoed in airway metabolomic data, and association with expression of immune genes in the host airway transcriptome[10]. Data from the COAST cohort ($n = 285$, Madison, WI, US) showed that a *Staphylococcus*-dominated trajectory in the first 6 months of life was associated with was associated with asthma throughout childhood and up to age 18[37]. The DORMICe study ($n = 159$, London, UK) found inverse associations between oropharyngeal *Granulicatella* from 9–12 m and *Prevotella* from 18–24 m and wheeze from 0–2 years, whereas *Neisseria* from 9–24 m was positively associated with wheeze[38]. In the STEPS study ($n = 704$, southwest Finland), researchers found that a persistent *Moraxella*-sparse profile from 2–13 months, compared with persistent *Moraxella* dominance, was associated with an increased risk of asthma by age 7[20].

These cohorts differ by selection criteria, countries of origin, timing, method of sampling, analytical approaches, and identified taxa. However, a common trend of association between early-life airway microbiota and wheezing or asthma remains. What is not clear from these results is whether specific bacterial taxa, bacterial functions, or host characteristics such as mucosal immune responses or other latent susceptibilities are key in forming this association. We lack a causal and mechanistic understanding which is sorely needed before research in this field can progress to the next step.

The most crucial question emerging from these studies is whether the bacteria are "to blame" for this association, by, e.g., instigating a trajectory of chronic inflammation, in which case one could envision targeted manipulation of the early-life airway microbiota as a future means of preventing or treating asthma, or whether the bacteria are differentially discovered in children at high risk for asthma due to inherent latent susceptibilities already present in early life, such as inadequate function or regulation of the mucosal immune system. Future studies should focus on in-depth characterizations of host-microbe interactions as well as developing translatable mechanistic model systems to assess causality and molecular mechanisms behind these birth cohort findings.

In conclusion, nasopharyngeal colonization at 1 month of age with *Streptococcus pneumoniae*, *Haemophilus influenzae*, and *Moraxella catarrhalis*, assessed by 16S sequencing and expressed as a pathogen score, was associated with the development of asthma by age 7 years, recapitulating earlier culture-based findings. The apparent frequency of colonization with *Veillonella* and *Prevotella* species in these naso-pharyngeal samples was lower than previous findings from the hypo-pharynx, which could be due to methodological differences, and did not associate with asthma in the current study, but associated positively with the pathogen score. Further clarification of dynamics in the temporospatial development of the upper airway microbiota and its relationship with host factors and later asthma is needed before preventive strategies can be conceived.

## Methods
### Study population and follow-up
We included samples obtained from infants in the COPSAC$_{2000}$ cohort, which consists of 411 children born to mothers with a history of asthma[29], in Copenhagen, Denmark. Exclusion criteria were severe congenital abnormalities, preterm birth <36 weeks of gestation, lung symptoms, or need for mechanical ventilation prior to enrollment at 1 month of age, between August 1998 and December 2001. Follow-up and data collection is still ongoing. The study was conducted in accordance with the guiding principles of the Declaration of Helsinki and was approved by the Local Ethics Committee (KF 01-289/96), and the Danish Data Protection Agency (2015-41-3696). Both parents gave written informed consent before enrollment. Children were followed in the clinical research unit at 1 month and 6 months of age, and every 6 months thereafter until the age of 7 years. In addition, children were

seen in the research unit instead of their family practitioner for acute visits in the event of any respiratory illness. The cohort also participated in the nested Prevention of Asthma in Childhood (PAC) Study; a randomized controlled trial of inhaled corticosteroid vs placebo at first wheezy episode for the prevention of asthma, which showed no effect[39].

### Nasopharyngeal swab sample sequencing
We collected nasopharyngeal swabs during the 4-week visit to the research clinic from September 1999 to December 2001 from asymptomatic neonates. The specimens were obtained from the posterior nasopharynx by placing an ENT cotton-tipped aluminum swab (Medical Wire & Equipment, Corsham, UK) for 15 s and rotating it two times before it was retracted and placed in 1.8 ml of SP4 mycoplasma transport medium[40–42]. The samples were kept at +4 °C until they were delivered to the laboratory the next day, where they were processed for mycoplasma culture as part of another protocol. The remaining sample material was immediately frozen at −80 °C and was not thawed before it was used for DNA extraction in 2021, where it was transported on dry ice to the lab. The hypopharyngeal aspirates were obtained on the same day as the nasopharyngeal swabs[5].

We extracted microbial genomic DNA from 250 µl nasopharyngeal saline using the epMotion® optimized NucleoSpin® 96 Soil DNA Isolation Kit (Macherey-Nagel, Düren, DE) in combination with the epMotion® robotic platform model (Eppendorf) according to the manufacturer's protocol. In order to prevent batch effects due to sampling, the samples were randomized during DNA extraction. Sterilized PBS solution and molecular-grade water were used as negative controls for DNA extraction and PCR, respectively, while a mock community was used as a positive control for PCR and sequencing. All operations were performed under aseptic conditions. The hypervariable V3-V4 regions of 16S rRNA genes were amplified with the modified broad primers 341F (5´-CCTAYGGGRBGCASCAG-3´) and Uni806R (5´-GGACTACNNGGGTATCTAAT-3´)[43] using Thermo Scientific™ Phusion High-Fidelity PCR Master Mix (Thermo Fisher Scientific, Waltham, MA, USA) in the first PCR. Amplification conditions for the first PCR were as follows: initial denaturation at 95°C for 1 min, 30 cycles of 95°C for 15 s, 56°C for 15 s, and 72°C for 30 s, followed by a final extension at 72°C for 5 min. In the second PCR, sequencing primers and adaptors were ligated to the amplicon library under the same amplification conditions but with only 15 cycles. The PCR products generated from both PCR steps were purified using Agencourt AMPure XP beads (Beckman Coulter Genomics, MA, USA) with the 96-well magnet stand. The purified 2nd PCR products were normalized by the SequalPrep™ Normalization Plate (96) Kit (Invitrogen Ltd., Paisley, UK), pooled in equimolar concentrations, concentrated using the DNA Clean & Concentrator™-5 Kit (Zymo Research, Irvine, CA, USA) and quantified by a NanoDrop (Thermo Fisher Scientific, Waltham, MA, USA). A final 4 nM pool containing all the samples in equimolar proportions was obtained for library preparation. Paired-end (2 × 300 bp) sequencing of the amplicon library was performed with the Illumina MiSeq System with Miseq reagent kit v3 (Illumina Inc., CA, USA), including 8% PhiX as an internal control. All measurements were taken from distinct samples. The lab work was conducted at the Section for Microbiology, Dept. of Biology, University of Copenhagen, Denmark.

### Bioinformatics analysis
Raw fastq files were demultiplexed using the MiSeq Controller software prior to downstream analysis. The primers and adaptors in sequencing reads were removed using Cutadapt[44]. The determination of amplicon sequence variants (ASVs) was performed on QIIME2 Core 2020.11 platform[45] using Amplicon Denoising Algorithm 2 (DADA2) analysis pipelines[46]. The resulting ASV sequences were annotated using the AnnotIEM pipeline (v.1.3), which combines sequence alignment against four databases: EzBioCloud[47] (r. 2018-05), NCBI[48]

(v. refseq 202), RDP[49] (v.11.5), and Silva[50] (v. 138SSU) followed by a high confidence selection of best probable annotation[51]. Species annotation was evaluated against a mock community in our positive sequencing controls, where it correctly annotated 19 out of 20 species and the last species could only be annotated to genus level.

To remove contamination from the human genome, we aligned representative ASV DNA sequences against the indexed reference human genome (GRCh38, Genome Reference Consortium human genome build 38) using Bowtie2 aligner (v.2.3.5) in --very-sensitive mode[52] and removed these mapped ASVs from the dataset.

Potential bacterial contaminants were removed using {decontam}[53] (v.1.16.0) by comparing real samples with negative controls in prevalence mode, any ASVs with more than .5 probability of being contaminants were removed, except those annotated as typical airway/skin microbiota: *Cutibacterium acnes*, *Dolosigranulum pigrum*, *Haemophilus influenzae*, *Moraxella catarrhalis*, *Staphylococcus aureus*, *Staphylococcus epidermidis*, *Staphylococcus hominis*, or *Streptococcus pneumoniae*. These species we assume were found in negative controls due to cross-contamination and/or barcode-drifting.

The COPSAC$_{2000}$ nasopharyngeal sequencing data were compared to both COPSAC$_{2000}$ hypopharyngeal cultures and COPSAC$_{2010}$ hypopharyngeal 16S sequencing (Local Ethics Committee: H-B-2008-093, Danish Data Protection Agency: 2015-41-3696), the methodologies of which are described in their respective publications[5,7], and reported below.

## COPSAC$_{2000}$ hypopharyngeal aspirate cultures

As described in the original publication[5], the hypopharyngeal aspirates from COPSAC$_{2000}$ were cultured for pathogenic bacteria: While the 1 month-old infants were sedated to allow lung-function testing, the doctors at the clinical research unit aspirated a sample from the hypopharyngeal region with a soft suction catheter passed through the nose into the hypopharynx. The samples were transported to the microbiology laboratories within 2 h after collection and cultured for bacteria with the use of standard methods for identification of *Streptococcus pneumoniae*, *Haemophilus influenzae*, *Moraxella catarrhalis*, *Staphylococcus aureus*, and *Streptococcus pyogenes*. The identification criteria for the species cultured were chosen before the study. The personnel in the clinical research unit were unaware of the results of the cultures. Because the infants were asymptomatic, the culture results were not reported to the parents, and the infants were not treated.

## COPSAC$_{2010}$ hypopharyngeal aspirate sequencing

As described in the original publication[7], the hypopharyngeal aspirates from COPSAC$_{2010}$ underwent 16S rRNA gene amplicon sequencing to determine the microbiota composition: Hypopharyngeal aspirates were obtained at age 1 month, using a soft suction catheter passed through the nose. We examined the airway microbiota using a 16S rRNA gene amplicon sequencing protocol targeting the V4 region: the aspirates were diluted in 1 ml sterile 0.9% NaCl and transported to the microbiological laboratory at Statens Serum Institut (Copenhagen, Denmark), where they were separated into 150 μl aliquots and stored at −80 °C until DNA extraction. DNA was extracted from 1 × 150 μl aliquots per sample using the PowerMag® Soil DNA Isolation Kit optimized for epMotion® (MO-BIO Laboratories Inc., Carlsberg, CA, USA) using the epMotion® robotic platform model (Eppendorf, Hamburg, Germany) under the manufacturer's protocol. All measurements were taken from distinct samples. At least one DNA extraction negative control was included in each 96-well plate, by adding 150 μl of molecular-grade water (Sigma-Aldrich, Merck, Germany) instead of a sample. DNA concentrations were determined using the Quant-iT™ PicoGreen® quantification system (Life Technologies, CA, USA). Extracted DNA was stored at −20 °C. 16S rRNA gene amplification of the hypervariable V4 region was performed over two PCR steps. First,

amplification of the 16S rRNA gene, using broad range primers (515 F (5′-GTGCCAGCMGCCGCGGTAA-3′) and 806 R (5′-GGACTAC HVGGGTWTCTAAT-3′)), with a reaction mixture consisting of 1× AccuPrime PCR Buffer II, 0.6 U AccuPrime Taq DNA Polymerase (Invitrogen, Life technologies, CA, USA), 0.5 μM primer 515 F, 0.5 μM primer 806 R, 2.0 μl template DNA, and molecular-grade water (Sigma-Aldrich, Merck, Germany) to a total volume of 20.0 μl per sample. Reactions were run in a 2720 thermal cycler (Applied Biosystems®, Life Technologies, CA, USA) according to the following cycling program: 2 min of denaturation at 94 °C, followed by 30 cycles of 20 s at 94 °C (denaturing), 30 s at 56 °C (annealing), and 40 s at 68 °C (elongation), with a final extension at 68 °C for 5 min. For each plate, a negative template-free control and a positive control containing 2.0 μl DNA from a known bacterial mock community (1.0 ng/μl; HM-782D, BEI Resources, VA, USA) were included. The PCR products were quantified using the Quant-iT™ PicoGreen® quantification system (Life Technologies, CA, USA), and samples with a PCR product concentration above 6.0 ng/μl were diluted to ~3.0–6.0 ng/μl prior to further analysis. Sequencing primers and adaptors were added to the amplicon products in the second PCR step: 2.0 μl of the diluted amplicons were mixed with a reaction solution consisting of 1× AccuPrime PCR Buffer II, 0.6 U AccuPrime Taq DNA Polymerase (Invitrogen, Life Technologies, CA, USA), 0.5 μM fusion forward and 0.5 μM fusion reverse primer, and molecular-grade water (Sigma-Aldrich, Merck, Germany) (total volume 20 μl). The PCR was run according to the cycling program above, except with a reduced cycling number of 15. The amplification products were purified with Agencourt AMPure XP Beads (Beckman Coulter Genomics, MA, USA) using 0.7× volume beads and quantified as described above. Equimolar amounts of the amplification products were pooled together in a single tube. The pooled DNA samples were concentrated using the DNA Clean & Concentrator™−5 Kit (Zymo Research, Irvine, CA, USA), and the concentrations were then determined using the Quant-iT™ High-Sensitivity DNA Assay Kit (Life Technologies). Paired-end sequencing of up to 192 samples was performed on the Illumina MiSeq System (Illumina Inc., CA, USA), including 1.0% PhiX as internal control, using MiSeq Reagent Kits v2 (Illumina Inc., CA, USA). Automated cluster generation and 250 paired-end sequencing with dual-index reads were performed. The resulting fastq files were then reanalyzed with the same pipeline as described above for the COPSAC$_{2000}$ nasopharyngeal swabs.

## Clinical investigator-diagnosed endpoints

Asthma-related clinical endpoints were diagnosed solely by COPSAC physicians according to a previously described quantitative symptom-based diagnostic algorithm[5,39,54,55], using structured diary cards of symptoms filled out daily by parents and validated by COPSAC physicians at visits to the research clinic. The longitudinal endpoints were defined as follows:

- *Wheezy episodes* as at least 3 days of continuous symptoms of coughing, wheezing, and/or breathlessness severely impacting the well-being of the child;
- *Recurrent wheeze* as at least five wheezy episodes within six months or four weeks of continuous symptoms;
- *Persistent wheeze/asthma* as a diagnosis of children with recurrent wheeze responding to a 3-month course of inhaled corticosteroids (ICS) and relapsing upon stopping treatment;
- *Exacerbations* as events of severe asthma-like symptoms requiring treatment with oral corticosteroids (OCS) or high dose ICS or requiring hospitalization.

## Covariates

Information on sex, ethnicity, delivery mode, gestational age at birth, birth weight, birth length, home address, siblings in the home, furred pets in the home, maternal smoking during pregnancy, breastfeeding, social circumstances, antibiotics during pregnancy and birth was

obtained by personal interviews with the parents at clinical visits and where applicable validated against the Danish Medical Birth Registry[56]. Living environment was derived based on home address using the CORINE satellite-based land cover database (2012 version) as previously described[31], categorized by five major land cover types for 100 × 100 m entities in a 3 km buffer and deriving urban vs rural living by partitioning around medoids clustering[57] of Euclidean distances.

## Statistical analysis

All statistical analyses were performed in R v. 4.2.1[58], and figures were prepared using {ggplot2} v.3.3.6[59]. Microbiome data were analyzed using the package {phyloseq} v.1.40.0[60]. Tables were prepared with {gtsummary} v.1.6.2[61]. Baseline variables were tested against having a sample included using Chi-Squared test (categorical, all expected counts ≥5), Fisher's exact test (categorical, any expected counts <5), or Wilcoxon rank-sum tests (continuous). Relative abundance values of individual taxa were compared between culture positive and negative groups using Wilcoxon tests and area under the curve (AUC) from receiving operator characteristic (ROC) curves. Youden's J index[62] was used to annotate ROC curves with potential cutoff values. Longitudinal analyses of wheeze/asthma were performed using Cox proportional hazards regression and quantified with Hazard Ratios (HR). Sex-specific results were obtained by stratification and tested using interaction terms in the full model. A pathogen score was derived by taking sample-wise sums of the relative abundance values of ASVs annotated as *Streptococcus pneumoniae*, *Haemophilus influenzae* and *Moraxella catarrhalis*, and log-transformed using the lowest nonzero value as pseudocount. To visualize the pathogen score, it was divided into tertiles and plotted using a Kaplan–Meier curve of time to persistent wheeze/asthma up to age 7 years. Differential abundance analysis was performed at species level (agglomerated from ASVs), using two methods: Cox regression with log-scaled species relative abundance values as predictors from the package {survival} v3.4–0 and {DESeq2}[63] v. 1.36.0, adjusted for log(library size) and sequencing run. These methods were tested and found robust to false positives in this dataset[64,65]. Alpha diversity (within-sample) was quantified using the Shannon diversity index and Faith's phylogenetic diversity index from the package {picante} v. 1.8.2[66]. Beta-diversity (between-sample) was quantified using the unweighted and weighted UniFrac metrics[67,68] and inference was calculated with the adonis2 PERMANOVA method from the R package {vegan} v.2.6–4[69]. Results for both alpha- and beta-diversity were adjusted for log(library size) and sequencing run. We analyzed the V3-V4 resolution within the *Moraxella* genus using RibDif[70]. We compared co-occurrences of taxa with Spearman and SparCC[71] correlations (as implemented in the R package {SpiecEasi} v. 1.1.2) and ordered axes using hierarchical clustering with the complete linkage method after converting correlation matrices to dissimilarity matrices with the function $d = (1−c)/2$. We obtained $p$ values for the SparCC correlations by bootstrapping with 1000 repeats.

No imputation of missing values was performed and a $p$ value below 0.05 was considered statistically significant throughout all analyses. All $p$ values were derived from two-sided tests. In the differential abundance analyses and pathogen score correlation taxa analyses, multiple comparisons were controlled using the Benjamini–Hochberg FDR correction[72].

## Reporting summary

Further information on research design is available in the Nature Portfolio Reporting Summary linked to this article.

## Data availability

Source data are provided with this paper. The 16S sequencing data generated in this project is deposited in the European Nucleotide Archive (ENA) with the accession no PRJEB58215 (https://www.ebi.ac.uk/ena/browser/view/PRJEB58215) in an anonymized form. The human genome used (GRCh38) is available from NCBI (ftp://ftp.ncbi.nlm.nih.gov/genomes/archive/old_genbank/Eukaryotes/vertebrates_mammals/Homo_sapiens/GRCh38/seqs_for_alignment_pipelines/GCA_000001405.15_GRCh38_no_alt_analysis_set.fna.bowtie_index.tar.gz). Individual-level personally identifiable clinical data from the children participating in the cohort cannot be made freely available, to protect the privacy of the participants and their families, in accordance with the Danish Data Protection Act and European Regulation 2016/679 of the European Parliament and of the Council (GDPR) that prohibit distribution even in pseudo-anonymized form. However, research collaborations are welcome, and data can be made available under a joint research collaboration by contacting the COPSAC Data Protection Officer (DPO), Ulrik Ralfkiaer, PhD (administration@dbac.dk). Requests will be answered within two weeks. Data use is restricted to purposes within childhood health and disease.

## Code availability

In addition to the publicly available tools detailed in the Methods, the study used an algorithm for taxonomic annotation (AnnotIEM), which is currently under preparation for publication as an open-access tool. Requests for access to AnnotIEM can be addressed to Avidan Neumann (avidan.neumann@uni-a.de). The reviewers had access to AnnotIEM during the revision process. R code used to generate figures and tables is available from the corresponding authors.

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

## Acknowledgements

We express our deepest gratitude to the children and families of the COPSAC cohort studies for all their support and commitment. We acknowledge and appreciate the unique efforts of the COPSAC research team. We thank Ulrik Ralfkiær for their graphical assistance. All funding received by COPSAC is listed on www.copsac.com. The Lundbeck Foundation (Grant no R16-A1694); The Ministry of Health (Grant no 903516); the Danish Council for Strategic Research (Grant no 0603-00280B) and The Capital Region Research Foundation have provided core support to the COPSAC research center. JS has received funding from the Danish Council for Independent Research (Grant no. 8045-00081B).

## Author contributions

The guarantor of the study integrity is J.S. J.T. prepared figures and tables. J.T., X.J.L., and J.S. drafted the manuscript. X.J.L. and S.P. sequenced the samples and generated ASVs. M.B. and A.U.N. developed the annotation pipeline. J.T., X.J.L., S.P., R.B.S., S.A.S., M.B., C.S.P., C.E.P., C.L.R., M.W., A.U.N., U.T., B.C., K.B., H.B., S.J.S., J.S. provided important intellectual input and contributed considerably to the analyses and interpretation of the data. J.T., X.J.L., S.P., R.B.S., S.A.S., M.B., C.S.P., C.E.P., C.L.R., M.W., A.U.N., U.T., B.C., K.B., H.B., S.J.S., J.S. guarantee that the accuracy and integrity of any part of the work have been appropriately investigated and resolved and all have approved the final version of the manuscript. The corresponding author had full access to the data and had final responsibility for the decision to submit it for publication. No honorarium, grant, or other form of payment was given to any of the authors to produce this manuscript.

## Competing interests

The authors declare no competing interests.
