## [Peer Review File · Nature Communications]

REVIEWER COMMENTS

Reviewer #1 (Remarks to the Author):

The leading authors have previously published findings from COPSAC cohorts, which were established by the late Dr. Bisgaard, and are considered some of the most well-characterized pediatric cohorts. In this submission, Dr. Thorsen and coworkers profile bacterial communities using 16S RNA (V3-V4 region) from nasopharynx swabs of 1month old children from the COPSAC2000 cohort. They compared these communities to hypopharyngeal aspirate diagnostic culture results (targeting three respiratory pathogens *H. influenzae*, *M. catarrhalis* and *S. pneumoniae*) from 244 children from the same cohort (data previously published Bisgaard et al., NEJM 2007). They report a strong correlation between culture results and sequenced communities when it comes to the detection of the three pathogenic species. A high pathogen score derived from the relative abundance of the three pathogens was associated with asthma, much like what has been previously reported based on culture. Bacterial communities in this study showed low diversity of oral anaerobes *Veillonella* and *Prevotella*, previously found in hypopharyngeal aspirates from 1mo children (COPSAC2010 – published Thorsen et al, Nat Comm 2019) to be associated with asthma – these genera were not found to be associated with asthma in this study – the authors attribute it to likely being due to sampling different sites of the airway.

Major comments:

After reading this cleanly put-together manuscript, I'm left perplexed about what novelty it adds to the field of airway microbiome in pediatric asthma. Correlation between culture and sequencing of the three respiratory pathogens had been established in this field, and their association with asthma development later in life is well established. The study is inadequately designed to address bacterial community composition between the hypopharynx and nasopharynx; as such, the conclusions related to the differences between the airway compartments are overstated. My disappointment also extends to the manuscript's narrative solely focused on the COPSAC cohorts – the narrative would benefit from expanding the scope to what is known about airway microbiome colonization in asthma.

Specific comments:

1. Nasopharyngeal swabs used for this study were stored in SP4 mycoplasma transport medium [line 199]; the samples were kept at 4C until delivery to the laboratory; they were then used for cultivation of mycoplasma – which means they were likely exposed to ambient temperature and air for at least a few hours before being stored at -80C. Can you confirm that these conditions

(specifically undergoing mycoplasma culture protocol) do not change bacterial profiles? Specifically in relation to anaerobic bacteria?

2. The median sequencing depth was 60723 with IQR 45384-79958 [lines 311-314]. Was the data normalized to one depth? Or did it undergo another form of normalization?

3. *S. aureus* is shown in Figure 2 with what looks like a good correlation to culture results, but there is no reference to it in the text – is there a reason why it was not included in the pathogen score? It has been linked to asthma outcomes in children, was there no association with *S.aureus* and any risk estimates? It at least deserves an explanation on why it was omitted from the pathogen score calculation (other than it was not included in Bisgaard et al 2007- but more recent evidence implicates it in asthma genesis). How do the results change if you include its abundance in an updated pathogen score?

4. Positive pathogen culture was a stronger predictor of asthma than detection of these by sequencing. Is that because sequencing is more sensitive than culture, which requires a higher relative abundance of these organisms in a sample? Have you looked at the distribution of samples by dominant organisms (not presence or absence but dominance of pathogens vs commensals)?

5. Shannon diversity was not associated with asthma [line 375] by age 7. What about phylogenetic diversity?

6. Overall microbial composition was not significantly different between children who developed asthma by age 7 and those who did not by weighted UniFrac [lines 376-378]. What about unweighted UniFrac?

7. Was bacterial diversity associated with the pathogen score? Or the relative abundance of specific pathogens or commensals?

8. “*Veillonella* and *Prevotella* are less common in the nasopharynx than hypopharynx, and do not associate with asthma” [lines 396-397]. This conclusion is an overstatement and may be misleading since the supporting data is derived from separate cohorts, where samples were not handled the same (COPSAC2000- samples are much older and were used for culture of mycoplasma before storage for sequencing); the sampling approach was different (aspirate vs swab). All these factors are known to affect the composition of the airway microbiome. Addressing similarities rather than differences between these two cohorts may be more convincing.

Reviewer #2 (Remarks to the Author):

This paper reports a novel analysis of airway microbiota in nasopharyngeal swabs of the same children in whom previously positive associations of airway pathogens cultured from early life hypopharyngeal aspirates with school-age asthma were found. The original findings were not replicated but a summation score of relative abundance values for 16S rRNA assessment of the previously associated pathogens was related to subsequent asthma.

The question arises why the previously described associations cannot be replicated. It remains unclear whether this is due to a lack of statistical power, selection by culturing, sampling location, or different 16S rRNA regions amplified. The authors might try to disentangle the different explanations. As it is now, too many parameters vary and do not allow for a sensible comparison and sound conclusions.

The main conclusion that “the nasopharyngeal microbiota, assessed using 16S sequencing, was strongly correlated with culturing of pathogenic airway bacteria from hypopharyngeal aspirates” is actually challenged by the quite modest AUROC values. Rather the discrepancy suggests that the cultures revealed a particular aspect that is “ignored” by 16s rRNA sequencing. This notion is supported by the statement in lines 349-352: “The association disappeared when repeating the analysis only among the children with no pathogenic bacteria detected by cultures (HR 0.93 [0.60;1.46], $p=0.44$, $n=188$) or in all children when adjusting for the presence of pathogenic bacteria (HR 1.06 [0.73;1.52], $p=0.76$)”. So culturing might reveal a distinct characteristic that is missed by direct amplification and sequencing.

The usage of a pathogen score is counterintuitive in a paper on the specificity of bacterial colonization. Moreover, it might be a proxy for host conditions rather than a summation of effects of pathogens. One may also perform sensitivity analyses leaving out one taxon at a time to quantify their relative contribution to the effect.

Figure 1 is very illustrative. However, one might also mention the risk profiles (family history) and the underlying intervention of COPSAC2010. Since COPSAC2010 is an intervention study, stratified or adjusted analyses are recommended. The selection of participants of observational studies and trials may vary. This should be considered when comparing the two different study types.

As suggested by Figures 1 and 4, COPSAC2010 is an essential part of this paper. So it should also be mentioned in the abstract.

Figure 2A is not very informative because AUROC is a global measure veiling potentially substantial differences between sensitivity and specificity. E.g. for *H. influenzae* sequencing is highly specific but hardly sensitive as illustrated by Fig. S1. Moreover, the scale of AUROC is prone to misinterpretation due to its offset of 0.5. One may replace Fig. 2A by Fig. S1, as it is not affected by the mentioned shortcomings.

There are no inverse correlations of asthma with sequenced taxa reported. They might be informative for protective taxa previously missed by culturing techniques.

There might be a cohort effect in the sense that lead pathogens might change over time. Is there any data available (e.g. from hospital microbiology records) that the spectrum of respiratory pathogens might have changed over the years?

REVIEWER COMMENTS

Reply: We thank the reviewers for the thorough evaluation and constructive suggestions for improving the manuscript.

Reviewer #1 (Remarks to the Author):

The leading authors have previously published findings from COPSAC cohorts, which were established by the late Dr. Bisgaard, and are considered some of the most well-characterized pediatric cohorts. In this submission, Dr. Thorsen and coworkers profile bacterial communities using 16S RNA (V3-V4 region) from nasopharynx swabs of 1month old children from the COPSAC2000 cohort. They compared these communities to hypopharyngeal aspirate diagnostic culture results (targeting three respiratory pathogens *H. influenzae*, *M. catarrhalis* and *S. pneumoniae*) from 244 children from the same cohort (data previously published Bisgaard et al., NEJM 2007). They report a strong correlation between culture results and sequenced communities when it comes to the detection of the three pathogenic species. A high pathogen score derived from the relative abundance of the three pathogens was associated with asthma, much like what has been previously reported based on culture. Bacterial communities in this study showed low diversity of oral anaerobes *Veillonella* and *Prevotella*, previously found in hypopharyngeal aspirates from 1mo children (COPSAC2010 – published Thorsen et al, Nat Comm 2019) to be associated with asthma – these genera were not found to be associated with asthma in this study – the authors attribute it to likely being due to sampling different sites of the airway.

Major comments:

Comment 1 (C1): After reading this cleanly put-together manuscript, I'm left perplexed about what novelty it adds to the field of airway microbiome in pediatric asthma. Correlation between culture and sequencing of the three respiratory pathogens had been established in this field, and their association with asthma development later in life is well established. The study is inadequately designed to address bacterial community composition between the hypopharynx and nasopharynx; as such, the conclusions related to the differences between the airway compartments are overstated. My disappointment also extends to the manuscript's narrative solely focused on the COPSAC cohorts – the narrative would benefit from expanding the scope to what is known about airway microbiome colonization in asthma.

Reply 1 (R1): We thank the reviewer for these spot-on comments and are grateful for the opportunity to remedy these shortcomings of the manuscript. The whole endeavor from our side –digging out these 20-years old samples hidden in our freezer and re-examining them using sequencing techniques, was indeed an effort to learn more about the “dark matter” of these famous old samples. The main research question was therefore whether additional knowledge could be gained regarding the old culture-based asthma associations. We conclude that the “dark matter” did not provide strong additional predictive power, and our findings can in that light be considered negative, but nevertheless important, as various newer studies have pointed to non-cultivable bacteria being important asthma-predictors. Our study was indeed not designed to compare hypopharynx and nasopharynx. While we addressed this limitation already in the previous manuscript version, the wording in the Conclusion and elsewhere did not adequately reflect that it was never our primary intention.

Rather, we had the opportunity to analyze these samples and had to first establish that they indeed were comparable with the old culture data. We have revised the manuscript to soften the wording regarding these differences but would be grateful to the reviewer and editor for pointing out any such places that we overlooked. In addition, we have underscored that our study was not designed to test such differences in detection; in the discussion:

“... However, other potential reasons include the numerous differences in sample collection, handling and storage – our study was not designed to infer the cause of such differences in observed abundances, but rather proved an opportunity to attempt to reconcile the apparent differences in asthma associations between the cohorts.”

Furthermore, we agree that the manuscript could benefit from “zooming out” to include other cohorts’ findings and to point forward to the challenges the field will be tackling in the coming years. This is now included as a substantial addition to the Discussion.

Specific comments:

C2: Nasopharyngeal swabs used for this study were stored in SP4 mycoplasma transport medium [line 199]; the samples were kept at 4C until delivery to the laboratory; they were then used for cultivation of mycoplasma – which means they were likely exposed to ambient temperature and air for at least a few hours before being stored at -80C. Can you confirm that these conditions (specifically undergoing mycoplasma culture protocol) do not change bacterial profiles? Specifically in relation to anaerobic bacteria?

R2: We thank the reviewer for highlighting this, and we agree with this sentiment – it cannot be ruled out, and may even be plausible, that this handling protocol, which differed from the one applied in the COPSAC2010 cohort, could contribute to differences between the two cohorts, and specifically in the reduction of detection of *Veillonella* and *Prevotella*, among other anaerobic taxa. We have now described this limitation in the Discussion section:

“Similarly, since the samples underwent culture for mycobacteria before freezing to -80°C, which could contribute to degradation of anaerobic bacteria like Veillonella and Prevotella. Notably, such skewness, like other technical biases, would equally affect all samples and thus not interfere with associations with clinical phenotypes. However, they could interfere with the comparisons made with other cohorts, including COPSAC2010.”

And in the interpretation regarding *Veillonella* and *Prevotella*,

“Another reason could be that the samples were not immediately frozen after collection.”

C3: The median sequencing depth was 60723 with IQR 45384-79958 [lines 311-314]. Was the data normalized to one depth? Or did it undergo another form of normalization?

R3: The data was not normalized to a single sequencing depth, but rather normalized in different ways as appropriate depending on the analysis. For most analyses, relative abundance normalization (Total sum scaling) was used, but e.g. in the differential abundance analysis, we used relative abundance normalization for the Cox regression and DESeq2’s standard quantile-type normalization for the DESeq2 analysis, which could contribute to part of the observed differences between the two methods. Where appropriate, we also included library size as a technical covariate used to adjust our analyses.

C4: *S. aureus* is shown in Figure 2 with what looks like a good correlation to culture results, but there is no reference to it in the text – is there a reason why it was not included in the pathogen score? It has been linked to asthma outcomes in children, was there no

association with *S.aureus* and any risk estimates? It at least deserves an explanation on why it was omitted from the pathogen score calculation (other than it was not included in Bisgaard et al 2007- but more recent evidence implicates it in asthma genesis). How do the results change if you include its abundance in an updated pathogen score?

R4: We did indeed base our analysis priorities on first recapitulating the culture-based findings from COPSAC2000 (where *S aureus* was cultured, but not associated with later asthma), with the new sequencing data before expanding the view to other taxa. Among the taxa examined in the differential abundance analysis was also *S aureus*, however it was not significant in neither the Cox regression nor the DESeq2 analysis. Therefore, including it in the pathogen score alongside the other three does not add anything to the analysis, but instead interferes with the signal from the three a priori chosen species (crude HR 1.25 [0.98;1.70], $p=0.15$). Please also refer to R8 where we show that *S. aureus* was actually inversely correlated with the pathogen score.

C5. Positive pathogen culture was a stronger predictor of asthma than detection of these by sequencing. Is that because sequencing is more sensitive than culture, which requires a higher relative abundance of these organisms in a sample? Have you looked at the distribution of samples by dominant organisms (not presence or absence but dominance of pathogens vs commensals)?

R5: We thank the reviewer for this suggestion of a dominance-based analysis. We have prepared a figure for the reviewer showing the distribution of the pathogen score before log transformation – in a relative abundance space. The vertical line represents the cutoff for pathogen dominance at 50%. As can be seen on the figure, the group of pathogen-dominated samples only comprise $n=14$. Despite this small group size, the signal towards asthma persists; crude HR 1.33 [1.12;1.58], $p=0.0014$. While the analysis certainly is of interest, the small group size ultimately limits its usefulness.

C6. Shannon diversity was not associated with asthma [line 375] by age 7. What about phylogenetic diversity?

R6: We thank the reviewer for this suggestion and have now included Faith's Phylogenetic Diversity to the analysis, since it provides additional and complementary information to the

existing alpha diversity analysis with the Shannon index. However, there was no significant association with asthma. The results have been updated:

“We quantified the diversity of the microbiome using the Shannon diversity index and Faith’s phylogenetic diversity, neither of which was associated with asthma (Cox regression, Shannon: HR 0.72 [0.42;1.22], p=0.22; Faith: HR 1.02 [0.88;1.18], p=0.78; n=285).”

C7. Overall microbial composition was not significantly different between children who developed asthma by age 7 and those who did not by weighted UniFrac [lines 376-378]. What about unweighted UniFrac?

R7: We agree with the reviewer it could be interesting to include this complementary distance metric and similar to the previous question on alpha diversity, we have expanded the results to include a comparison using the unweighted UniFrac metric.

“We compared the overall microbiome composition of children who developed asthma by age 7 with those that did not, which were similar (PERMANOVA, unweighted UniFrac, F=1.21, p=0.17; log weighted UniFrac, F=1.17, p=0.25; n=221).”

C8. Was bacterial diversity associated with the pathogen score? Or the relative abundance of specific pathogens or commensals?

R8: We analyzed the Shannon diversity index and Faith’s phylogenetic diversity, in line with the update in R6. We found that indeed, as suggested by the reviewer, both metrics were associated with the pathogen score (see plots including results from linear models and Spearman’s correlation tests below). Furthermore, we analyzed these diversity metrics in relation to the same subset of taxa above the cutoff for the differential abundance analysis (Fig. 3). We found that most taxa were positively associated with both Shannon and Faith diversity metrics, this was not particularly different between our three pathogens of interest (marked with color) and other taxa. In particular, diversity was inversely associated with *Staphylococcus aureus*, which makes a lot of sense, since this is by far the taxon with highest relative abundance, thereby numerically “suppressing” the relative abundance of other taxa, which will also “suppress” diversity metrics. In the lollipop plots below, one star is $p < 0.05$, two stars $q < 0.05$.

Finally, to wrap up this interesting analysis, we compared the pathogen score with each taxon (which is related to the *Veillonella/Prevotella*-focused figure S5). Here, we found that, unsurprisingly, the three pathogens on which it was based exhibited the strongest correlations. However, the 4th and 5th strongest were seen from two *Veillonella* species, hinting at a potential shared subcommunity structure as seen in Figure S5. Note that *Prevotella* was too rare to be above the threshold for this analysis. Again, this was in opposite direction with *Staphylococcus spp.*

To summarize, while the pathogen score is indeed associated with diversity, the two metrics express different traits in terms of taxa composition, where the pathogen score is much more specific for the three pathogens and their related co-colonizing taxa.

Linear model, Shannon diversity vs log relative abundance, 35 taxa, n = 285

C9. “*Veillonella* and *Prevotella* are less common in the nasopharynx than hypopharynx, and do not associate with asthma” [lines 396-397]. This conclusion is an overstatement and may be misleading since the supporting data is derived from separate cohorts, where samples were not handled the same (COPSAC2000- samples are much older and were used for culture of mycoplasma before storage for sequencing); the sampling approach was different (aspirate vs swab). All these factors are known to affect the composition of the airway microbiome. Addressing similarities rather than differences between these two cohorts may be more convincing.

R9: As noted in R1 and R2, we completely agree with this assessment and have adjusted the wording accordingly throughout the manuscript. Our study was not designed to compare nasopharynx and hypopharynx directly, rather; we had to use the samples/data at our disposal with the inherent limitations stemming from that. Please refer to R1 and R2 for related answers.

Reviewer #2 (Remarks to the Author):

This paper reports a novel analysis of airway microbiota in nasopharyngeal swabs of the same children in whom previously positive associations of airway pathogens cultured from early life hypopharyngeal aspirates with school-age asthma were found. The original findings were not replicated but a summation score of relative abundance values for 16S rRNA assessment of the previously associated pathogens was related to subsequent asthma.

C10: The question arises why the previously described associations cannot be replicated. It remains unclear whether this is due to a lack of statistical power, selection by culturing, sampling location, or different 16S rRNA regions amplified. The authors might try to disentangle the different explanations. As it is now, too many parameters vary and do not allow for a sensible comparison and sound conclusions.

R10: As outlined in figure 1, there are several previous findings that we build on in this figure. The first one is the culture results from COPSAC2000 and asthma. We conclude that these findings were recapitulated with the sequencing data, but we do not consider this a replication, since the cohort is the same. The second result is the findings from COPSAC2010 based on sequencing data concerning *Veillonella*, *Prevotella*, and asthma. This finding we were not able to directly replicate, seemingly due to low detection in the COPSAC2000 samples. Here, we have now expanded the description of the differences that may contribute to the reduced detection of these taxa, please refer to R1, R2, and R9 for further details.

C11: The main conclusion that “the nasopharyngeal microbiota, assessed using 16S sequencing, was strongly correlated with culturing of pathogenic airway bacteria from hypopharyngeal aspirates” is actually challenged by the quite modest AUROC values. Rather the discrepancy suggests that the cultures revealed a particular aspect that is “ignored” by 16s rRNA sequencing. This notion is supported the statement in lines 349-352: “The association disappeared when repeating the analysis only among the children with no pathogenic bacteria detected by cultures (HR 0.93 [0.60;1.46], $p=0.44$, $n=188$) or in all children when adjusting for the presence of pathogenic bacteria (HR 1.06 [0.73;1.52], $p=0.76$)”. So culturing might reveal a distinct characteristic that is missed by direct amplification and sequencing.

R11: We mostly agree with this comment from the reviewer, especially the interesting phenomenon that it seems that the culture data capture something unique which is not fully covered in the sequencing data. As the reviewer also mentions in C10, pertaining to the comparison of nasopharyngeal sequencing and hypopharyngeal culture, we acknowledge the limitation that at least 2 important factors vary together: sampling location and detection technique, which precludes us from concluding which is more important. When taking this into account, we however think that the associations are indeed quite strong, since both culturing and 16S sequencing are imperfect methods, which adds noise to the system, we would not expect AUC values anywhere close to 1. While not directly comparable, in a previous study we also found imperfect agreement of culturing and sequencing of both airway and fecal samples (Gupta et al, Communications Biology 2019, doi 10.1038/s42003-019-0540-1) – of note, this was a comparison with both techniques done on the same sample set. So, it is important to interpret these numeric estimates in light of what can realistically be expected. In this paper, the main motivation for the analysis is to show that these detection methods show agreement before it is reasonable to move on to the main point of the paper, which is the association with asthma.

C12: The usage of a pathogen score is counterintuitive in a paper on the specificity of bacterial colonization. Moreover, it might be a proxy for host conditions rather than a summation of effects of pathogens. One may also perform sensitivity analyses leaving out one taxon at a time to quantify their relative contribution to the effect.

R12: We agree with the reviewer that this colonization pattern may very well be a host-associated factor. As mentioned in the limitations, we cannot infer any directionality or causality from these observational data. This applies regardless of choice of detection method (culturing vs sequencing) and any data analysis approaches which can be applied. Based on this comment from the reviewer, which is partially echoed in C1, we have found it prudent to elaborate on this aspect in the discussion:

“What is not clear from these results is whether specific bacterial taxa, bacterial functions, or host characteristics such as mucosal immune responses or another latent susceptibility are key in forming this association. We lack a causal and mechanistic understanding which is sorely needed before research in this field can progress to the next step.

The most crucial question here is whether the bacteria are “to blame” for this association, by e.g. initiating a trajectory of chronic inflammation, in which case one could envision targeted manipulation of the early-life airway microbiota as a future means of preventing or treating asthma, or whether the bacteria are differentially discovered in children at high risk for asthma due to inherent latent susceptibilities already present in early life.”

As suggested by the reviewer, we have performed a sensitivity analysis leaving out each of the three pathogenic species in turn, see below. We conclude that no single species seems to be driving the association, which is mirrored by the individual species associations with asthma presented in Figure 3, in which all three species show the same directionality of association.

Model	HR	std.error	statistic	p.value	conf.low	conf.high	n	n_missing
Full pathogen score	1.50	0.15	2.72	0.00658	1.12	2.01	285	0
S. pneumoniae left out	1.53	0.12	3.49	0.00049	1.20	1.94	285	0
H. influenzae left out	1.28	0.14	1.74	0.08272	0.97	1.70	285	0
M. catarrhalis left out	1.34	0.14	2.07	0.03814	1.02	1.77	285	0

C13: Figure 1 is very illustrative. However, one might also mention the risk profiles (family history) and the underlying intervention of COPSAC2010. Since COPSAC2010 is an intervention study, stratified or adjusted analyses are recommended. The selection of participants of observational studies and trials may vary. This should be considered when comparing the two different study types.

C13: We thank the reviewer for these excellent suggestions and have updated figure 1 to include this information. Since the novel analysis in the current manuscript is performed in the older COPSAC2000 cohort, there were no pregnancy interventions. In addition to the updated fig 1, we highlighted the difference in selection criteria between the two COPSAC cohorts in the discussion:

“First, COPSAC₂₀₀₀ is a high-risk cohort where all the mothers had a history of doctor-diagnosed asthma¹⁴. In contrast, the COPSAC₂₀₁₀ cohort was unselected by design and had a rate of 30% self-reported asthma, which was higher than those who were invited but declined to participate⁵⁸.”

C14: As suggested by Figures 1 and 4, COPSAC2010 is an essential part of this paper. So it should also be mentioned in the abstract.

R14: We agree and have now included COPSAC2010 in the abstract. Note the abstract is now completely reformatted to comply with the journal style.

C15: Figure 2A is not very informative because AUROC is a global measure veiling potentially substantial differences between sensitivity and specificity. E.g. for H. influenzae sequencing is highly specific but hardly sensitive as illustrated by Fig. S1. Moreover, the scale of AUROC is prone to misinterpretation due to its offset of 0.5. One may replace Fig. 2A by Fig. S1, as it is not affected by the mentioned shortcomings.

C15: We agree with many of these technical assessments on the interpretation of ROC curves presented here. We considered the suggested changes to the figures and have decided to change the figures, but in a slightly different way. We have removed the AUC values printed to avoid such misinterpretation, and instead written the median + interquartile range of relative abundances in each group in the text. The reasoning behind this, which is related to R1, R2, and R9, is that the aim of the analysis is not to infer differences between the nasopharyngeal and hypopharyngeal microbiome. Rather, it is a necessary first step to show that despite the difference in sampling, we are still picking up a closely related signal, which is a prerequisite for the main objective – studying the association between bacterial colonization and asthma. We would therefore like to focus less on the “prediction” aspect – we here instead show that these samples represent distinct but related niches.

The more technical considerations of sensitivity, specificity and AUC values are, in our opinion, better relegated to a supplemental figure, as to not throw off the reader before reaching the main objectives of the study.

C16: There are no inverse correlations of asthma with sequenced taxa reported. They might be informative for protective taxa previously missed by culturing techniques.

C16: We agree with the reviewer that any such protective taxa would be of immense interest. The way our analysis is set up, we would be able to identify any such taxa if they displayed inverse associations with asthma. However, none turned out significant in our analysis, similar to the results from the COPSAC2010 cohort, where we likewise only detected positive associations. This may be airway specific and is in contrast to findings from the early life gut microbiota, where many taxa seem protective of later asthma development.

C17: There might be a cohort effect in the sense that lead pathogens might change over time. Is there any data available (e.g. from hospital microbiology records) that the spectrum of respiratory pathogens might have changed over the years?

R17: First of all, we agree with this assessment from the reviewer.

The reviewer is indeed correct that there is an extensive surveillance of pathogens in the Danish hospital system. There are public reports such as the DANMAP (https://www.danmap.org/-/media/sites/danmap/downloads/reports/2021/danmap_2021_version-1.pdf) and an online dashboard (<https://statistik.ssi.dk/>), which unfortunately are not optimal for this question, since they focus mostly on antimicrobial resistance and surveillance of invasive bacterial infections, such as pneumococcal meningitis. The internal hospital microbiology records system, MiBa, is not open to research currently pending some unsolved legal barriers (<https://miba.ssi.dk/forskningsbetjening>), except data extracted for the official surveillance reports. However, the point still stands and may even extend to changes in strains/serotypes of pathogens, which is probably even more fine-grained than what the system captures. We

have briefly touched on this in the discussion, where we speculate that eg. pneumococcal strains may have changed in response to the infant pneumococcal vaccine, which differed between the two cohorts. However, in the absence of more concrete data, it is limited what we can conclude apart from posing this hypothesis. We have now added a line that similar changes may have occurred over time for other taxa.

*“Fourth, with regard to *S. pneumoniae*, Denmark introduced the pneumococcal vaccination as part of the childhood vaccination programme in 2007⁵⁹. While the infants in the COPSAC₂₀₁₀ study were not yet vaccinated themselves at the time of sampling, the high child vaccination rates in the population (>95% in Denmark⁶⁰) can influence the *S. pneumoniae* strains in the community⁶¹ and potentially affect associations with asthma. Similar changes may have occurred over time for other taxa.”*

REVIEWER COMMENTS

Reviewer #1 (Remarks to the Author):

Thank you for the opportunity to review a revised manuscript entitled “The airway microbiota of neonates colonized with asthma-associated pathogenic bacteria.” by Dr Thorsen and colleagues.

I appreciate the effort put into revising the earlier version of the manuscript; however, my major criticisms of this work were dismissed; therefore, my original assessment still stands.

This manuscript does not provide any novel findings which advance the field of airway microbiome in pediatric asthma for the following reasons:

- The primary finding of a correlation between microbial culture and detection of three well-established asthma genic respiratory pathogens by sequencing has been previously published and is an accepted fact in the community. Here is a more recent example: Toivonen et al., Pediatrics (2020) <https://doi.org/10.1542/peds.2020-0421>
- The study design is flawed. Since the swab samples from COPSAC2000 sequenced in this study were used for Mycoplasma culture before storage and subsequent sequencing in this study - therefore are not representative of the nasopharynx at the time of collection. Clinical microbiome investigations are moving away from sequencing samples from “cohorts of convenience” for the sake of reporting sequenced data (once a novelty) to well-thought-out clinical studies not restricted by inadequately collected/handled or processed samples, which significantly undermines the current study's findings.
- The study was inadequately designed to examine differences in bacterial composition in distinct compartments of the airway; as such the comparison to COPSAC2010 Hypopharynx aspirate is of limited value and distracts from the main objective of the study (as defined by the authors themselves).
- Although the authors addressed some of my previous concerns in their direct responses, they did not attempt to restructure the analysis or the paper structure to uncover a novel clinically relevant observation that moves the field forward, which was a major criticism of the original submission.

Reviewer #2 (Remarks to the Author):

I thank the authors for their open responses. The main message of the paper is much clearer now: the most important asthma pathogens (*Haemophilus influenzae*, *Streptococcus pneumoniae*, and *Moraxella catarrhalis*) are found in different studies by different methods at different sampling locations. This is reassuring but the novelty is somewhat limited given other publications on this topic since the legendary NEJM 2007 paper.

In addition, this finding does not answer the question on discrepant findings as stated in the introduction: “it has remained unclear whether the differences in asthma-associated bacteria between these studies were due to differences in sampling, the applied detection technique (culturing vs sequencing), or simply due to inherent cohort differences.”

Obviously, the study design was not set up to answer this question and it would be unfair to rate studies performed 10 or 20 years ago against the current state of the art. However, as the mentioned question is a central element of this paper it should receive a more definitive answer.

For the two COPSAC studies, the current findings reconcile previous discrepancies with respect to the 2007 pathogens but the prominent role of *Veillonella* and *Prevotella* in COPSAC2010 remains enigmatic. The current paper should give a more definitive answer to this question. Was this due to an artefact? Are said taxa bystanders of the 2007 pathogens? This is suggested by the correlations reported at the end of the current results section but an adjustment for the correlation in the previous associations with asthma is missing.

Minor comments:

Line 154 (“while the pathogen colonization was associated with hospitalization for wheeze”): This statement should be amended by an odds ratio.

For *M. lincolnii* to be a new candidate since the NEJM 2007 publication one should clearly state that the new candidate is not just a product of an updated version of the taxonomic assignment. The wording in lines 173-5 is probably not clear enough.

Lines 218/9: I would also mention that adjustment for confounders did not produce results substantially different from the raw estimates since adjustment for many measured confounders can amplify residual confounder bias.

Line 304: “nuances” seems inappropriate given the substantial differences just listed.

Unfortunately the main document contains the main text twice but not the supplementary figures. From the responses I understand that there were no major changes in the supplement. Therefore I would not insist on assessing the supplement.

REVIEWER COMMENTS

Reviewer #1 (Remarks to the Author):

Comment 1: Thank you for the opportunity to review a revised manuscript entitled “The airway microbiota of neonates colonized with asthma-associated pathogenic bacteria.” by Dr Thorsen and colleagues.

I appreciate the effort put into revising the earlier version of the manuscript; however, my major criticisms of this work were dismissed; therefore, my original assessment still stands. This manuscript does not provide any novel findings which advance the field of airway microbiome in pediatric asthma for the following reasons:

The primary finding of a correlation between microbial culture and detection of three well-established asthma genic respiratory pathogens by sequencing has been previously published and is an accepted fact in the community. Here is a more recent example:

Toivonen et al., Pediatrics (2020) <https://doi.org/10.1542/peds.2020-0421>

R1: Thank you for once again evaluating our manuscript. The major concerns raised by the reviewer in the previous review were definitely not dismissed, but rather led to a large revision and a more precise and strengthened manuscript, for which we are grateful. As noted in the previous response letter, the comparison of culture and sequencing is not the primary finding of our study but only included as an intermediate result – in essence a prerequisite for the primary objective: To recapitulate the pathogen-asthma association using sequencing data and to look for potential novel taxa contributing to this phenomenon. Before we could examine the association with asthma, we first needed to establish that the two methods (culture and sequencing) were in agreement, also due to the differences in sampling.

We acknowledge that this misunderstanding about the study’s focus is likely due to imprecise wording in the introduction, which we have now amended, see R3.

The interesting study mentioned by the reviewer is cited in our manuscript, due to its findings of an association between longitudinal microbiota profiles and asthma (Ref 37).

C2: The study design is flawed. Since the swab samples from COPSAC2000 sequenced in this study were used for Mycoplasma culture before storage and subsequent sequencing in this study -therefore are not representative of the nasopharynx at the time of collection. Clinical microbiome investigations are moving away from sequencing samples from “cohorts of convenience” for the sake of reporting sequenced data (once a novelty) to well-thought-out clinical studies not restricted by inadequately collected/handled or processed samples, which significantly undermines the current study's findings.

R2: This comment is a continuation of the above misunderstanding of the aim of the study, which was not to compare the two microbiological compartments/techniques but to reassess and expand the association with asthma development using a culture independent technique. While the swab type and handling might bias our ability to claim a precise microbiota composition, it was not biased towards the asthma endpoint. The strong recapitulation of the association between pathogenic bacteria and asthma shows that the swabs were indeed still useful despite any delays before freezing and the 20 years of storage they were subjected to. See also below.

C3: The study was inadequately designed to examine differences in bacterial composition in

distinct compartments of the airway; as such the comparison to COPSAC2010 Hypopharynx aspirate is of limited value and distracts from the main objective of the study (as defined by the authors themselves).

R3: Please refer to responses above. We assume the reviewer is referring to this section, from the Introduction:

“Our aim was to use this data to examine whether the pathogen culture results could be re-established using sequencing data from these anatomically distinct but adjacently collected samples. Further, we examine whether any other taxa associating with asthma could be identified using the more sensitive sequencing method – in particular a possible replication of the COPSAC₂₀₁₀ Veillonella and Prevotella associations.”

We agree that this could be misunderstood as if the aim was to compare the different airway compartments. We are grateful for the chance to make this clearer, in particular what the word “results” in the first sentence refers to. We have now changed this in the hope that such misunderstanding can be avoided for new readers:

“Our aim was to use this data to examine whether the pathogen culture association with asthma could be re-established using sequencing data from these anatomically distinct but adjacently collected samples.”

We have also prefaced the relevant section of the Results:

“First, we compared the sequenced nasopharyngeal swabs to the hypopharyngeal aspirate culture results originally reported to be associated with asthma by age 5⁵, in order to ensure that there was agreement between these two sample sets and methods before comparing their associations with asthma.”

And finally, before progressing to the asthma associations in the Results,

*“Having established that these two samples were to an extent comparable, we progressed to our main aim of investigating if the association between neonatal pathogen colonization with *S. pneumoniae*, *H. influenzae*, and *M. catarrhalis* and asthma⁵ could be recapitulated with the sequencing data [...]”*

C4: Although the authors addressed some of my previous concerns in their direct responses, they did not attempt to restructure the analysis or the paper structure to uncover a novel clinically relevant observation that moves the field forward, which was a major criticism of the original submission.

R4: We indeed amended and expanded the manuscript in multiple ways based on pertinent comments from the reviewers in the previous responses. Keeping in mind our focus of gaining a better understanding of the association between neonatal bacterial colonization and asthma, we believe that our results here are indeed highly clinically relevant – establishing that these pathogenic species remain associated with asthma despite differences in sampling location, handling, and detection method. Furthermore, we were able to correlate these pathogens with *Veillonella* and *Prevotella*, which have been further analyzed and corroborated on in the current revision (please refer to R7). Our ultimate goal is to translate an understanding of this phenomenon into new avenues for asthma prevention in early life.

Reviewer #2 (Remarks to the Author):

C5: I thank the authors for their open responses. The main message of the paper is much

clearer now: the most important asthma pathogens (*Haemophilus influenzae*, *Streptococcus pneumoniae*, and *Moraxella catarrhalis*) are found in different studies by different methods at different sampling locations. This is reassuring but the novelty is somewhat limited given other publications on this topic since the legendary NEJM 2007 paper.

R5: We thank the reviewer for this assessment. We agree that the novelty in the study does not lie in simply showing an association between pathogen colonization and asthma. Rather, it lies in our finding that this phenomenon seems to be sufficiently described by these three pathogens alone. Surprisingly, we found no other taxa to be individually significant, as we might have expected based on newer sequencing-based findings in COPSAC2010 and other cohorts. Furthermore, it seems that the presence of these three pathogens could be part of the same latent pattern of *Veillonella* and *Prevotella* by virtue of their positive correlation in our data. See also R7.

C6: In addition, this finding does not answer the question on discrepant findings as stated in the introduction: “it has remained unclear whether the differences in asthma-associated bacteria between these studies were due to differences in sampling, the applied detection technique (culturing vs sequencing), or simply due to inherent cohort differences.”

Obviously, the study design was not set up to answer this question and it would be unfair to rate studies performed 10 or 20 years ago against the current state of the art. However, as the mentioned question is a central element of this paper it should receive a more definitive answer.

R6: We agree that this could be concluded and conveyed in a clearer manner. While our data does not allow us to fully reconcile these differences, we can conclude that the pathogen-asthma association did not critically depend on sampling location nor detection technique. This underscores the lack of association for the three pathogens in the 2010 cohort, and highlights that the observed differences must be due to inherent cohort differences, which we provide several examples for. This has now been clarified in the discussion:

“We can establish based on these results that the association between pathogenic bacteria and asthma in COPSAC₂₀₀₀ did not critically depend on sampling location and method nor detection method, which extends to the unresolved differences between the asthma-associated bacteria between the two COPSAC cohort studies listed in the introduction – and points to inherent cohort characteristics as the most important factor behind these differences.”

C7: For the two COPSAC studies, the current findings reconcile previous discrepancies with respect to the 2007 pathogens but the prominent role of *Veillonella* and *Prevotella* in COPSAC2010 remains enigmatic. The current paper should give a more definitive answer to this question. Was this due to an artefact? Are said taxa bystanders of the 2007 pathogens? This is suggested by the correlations reported at the end of the current results section but an adjustment for the correlation in the previous associations with asthma is missing.

R7: This is indeed an unresolved and even enigmatic question that we share a keen interest in trying to understand. The correlation results that the reviewer refers to (Supplementary Fig. 5) showed that the pathogen score was positively associated with the relative abundance of *Veillonella*, *Prevotella* and several of their species, but did not further explore this phenomenon. We have now expanded this analysis to compare these relationships against the backdrop of the rest of the microbial community, included as a new Supplemental Fig. 6.

Here, we show some quite interesting new details. Using two complementary correlation metrics (spearman + SparCC), we show that not only are *Veillonella* and *Prevotella* correlated with the pathogenic genera, they also seem to form a subcommunity structure as we have previously hypothesized. This clustering is shown in the heatmap, which was ordered using hierarchical clustering based on each correlation metric. Furthermore, we show that *Veillonella* and *Prevotella* abundances also are among the taxa strongest correlated with the pathogen score, and by stratified analyses that the correlation for *Prevotella* is likely underestimated due to undersampling.

We cannot justify making strong claims as to which of these taxonomic entities are true vs which may be bystanders on the basis of these observational data, we can merely show and report this interesting apparent link between them, which we speculate has relevance for the reported asthma associations and may even be different markers for the same underlying phenomenon.

Pertaining to the idea of an adjusted analysis, we have provided the results in a table below, however we are not convinced that such an adjusted analysis can help disentangle the relationship between these bacteria, for the following reasons: First, if these bacteria (pathogens + V/P) are indeed representatives of the same latent subcommunity, such an adjustment would detract from their joint signal and be an overadjustment. Second, since in the overall analysis the pathogen score was significant and *Veillonella/Prevotella* was not, there doesn't seem to be a sufficiently strong signal for them to attempt adjustment. With these caveats in mind, we see in the table that when mutually adjusted, the estimate for the pathogen score remains unchanged while the estimate for V/P attenuates but remains non-significant. We do not think these results add sufficient insight to warrant inclusion in the paper, but are of course open to counter-arguments.

We have now described this new figure at the end of the Results section.

Crude (as previously reported in manuscript)	HR	95% CI		p-value
		Lower	Upper	
Veillonella/Prevotella (presence vs absence)	1.27	0.73	2.23	0.40
Pathogen score (per SD)	1.50	1.12	2.01	0.0066
Mutually adjusted				
Veillonella/Prevotella (presence vs absence)	1.09	0.62	1.92	0.77
Pathogen score (per SD)	1.49	1.11	2.01	0.0085

Minor comments:

C8: Line 154 (“while the pathogen colonization was associated with hospitalization for wheeze”): This statement should be amended by an odds ratio.

R8: We have now added a Hazard Ratio:

“Of note, while the pathogen colonization was associated with hospitalization for wheeze (by age 5, HR 3.88 [1,92;7.86], p=0.00016, n=319), the pathogen score only showed a non-significant trend.”

C9: For *M. lincolnii* to be a new candidate since the NEJM 2007 publication one should clearly state that the new candidate is not just a product of an updated version of the

taxonomic assignment. The wording in lines 173-5 is probably not clear enough.

R9: We agree that this is an important point to communicate clearly, and have now added some context for this:

“M. catarrhalis and M. lincolnii are two distinct species, and to confirm that their 16S rRNA gene sequences are indeed different enough to confidently separate them in our data, we investigated their genetic similarity. We found them distinctly identifiable in the V3-V4 region used in this study based on 16S rRNA genes in reference genomes (Supplementary Fig. 4).”

C10: Lines 218/9: I would also mention that adjustment for confounders did not produce results substantially different from the raw estimates since adjustment for many measured confounders can amplify residual confounder bias.

R10: We have now mentioned this.

C11: Line 304: “nuances” seems inappropriate given the substantial differences just listed.

R11: We agree and have rephrased this: “Despite these differences, ...”

C12: Unfortunately the main document contains the main text twice but not the supplementary figures. From the responses I understand that there were no major changes in the supplement. Therefore I would not insist on assessing the supplement.

R12: We apologize – the supplement was omitted from the submission by mistake. There were only very minor formatting changes between the two versions (eg. “Supplemental fig 1” instead of “Figure S1”).

Note that in this submission, we have now added Supplemental Fig. 6, as detailed in R7.

REVIEWERS' COMMENTS

Reviewer #2 (Remarks to the Author):

I thank the authors for their clarifications. From R5 I understand that the main result of the paper is that the association between pathogen colonization and asthma is “sufficiently described by these three pathogens alone”. I appreciate the usage of “sufficiently” (instead of “exclusively”) because the study does not have the statistical power to demonstrate that no other pathogens are involved. However, the presentation of Veillonella and Prevotella in Fig. 4c suggests independent effects by those two genera, which is misleading and might be understood as contradicting the main result.

In this context, I appreciate the mutually adjusted analyses presented in R7. Though the presence/absence of Veillonella/Prevotella is correlated with the pathogen score, there is no evidence of collinearity or, to use the authors’ words, “overadjustment”. Anyway, the change in the estimate is suggestive of confounding by the pathogen score. Therefore I recommend replacing figure 4c by a forest plot showing the raw and mutually adjusted models presented in R7. I do not consider this a contradiction to the earlier analysis presented in Thorsen et al, Nat Comm 2019. Rather I share the authors’ perception that they might have captured the same signal in both cohorts, but in COPSAC 2010 by proxies of the three pathogens. Therefore, I consider the new Supplementary Fig. 6 very helpful; panels a and b might be presented in the main manuscript.

I share the other reviewer’s concerns that the study design is not suitable for comparing sampling sources and detection methods. However, I acknowledge that both COPSAC studies were not set up for such comparisons. Therefore, I recommend emphasizing that (1) such comparisons have been covered by other more suited studies and (2) the currently presented analyses are secondary analyses of studies with different original aims. The latter, however, involve the relation of airway pathogens and asthma development, which is in the focus of the current manuscript.

With respect to response R6, I suggest shifting the focus from the difference between the studies towards the common finding of the pathogen cluster. In my opinion, consistency between studies can be demonstrated without explaining the differences between them. Besides, the studies are underpowered to exclude differences. Insofar, the statement “We can establish based on these results that the association between pathogenic bacteria and asthma in COPSAC2000 did not critically depend on sampling location and method nor detection method” is too strong.

The argument in lines 226-229 is not logical as it mixes up the issues of temporality and confounding. Besides I would not consider pharmacological treatment in a cross-sectional study more problematic than the intervention performed in COPSAC2010.

Minor:

It might be easier for the reader to state in the figure legends which study is the basis of the analyses.

The use of “abundance” in the plural is jargon and might be replaced by “abundance values”.

REVIEWERS' COMMENTS

We thank the reviewer for agreeing to assess our revised manuscript and for highlighting some relevant points in their assessment. Replies are given to each comment in blue font.

Reviewer #2 (Remarks to the Author):

Comment 1: I thank the authors for their clarifications. From R5 I understand that the main result of the paper is that the association between pathogen colonization and asthma is “sufficiently described by these three pathogens alone”. I appreciate the usage of “sufficiently” (instead of “exclusively”) because the study does not have the statistical power to demonstrate that no other pathogens are involved. However, the presentation of Veillonella and Prevotella in Fig. 4c suggests independent effects by those two genera, which is misleading and might be understood as contradicting the main result.

Reply 1: We agree with this point – we cannot rule out smaller effects from other bacteria. Regarding fig 4c, see R2.

C2: In this context, I appreciate the mutually adjusted analyses presented in R7. Though the presence/absence of Veillonella/Prevotella is correlated with the pathogen score, there is no evidence of collinearity or, to use the authors’ words, “overadjustment”. Anyway, the change in the estimate is suggestive of confounding by the pathogen score. Therefore I recommend replacing figure 4c by a forest plot showing the raw and mutually adjusted models presented in R7. I do not consider this a contradiction to the earlier analysis presented in Thorsen et al, Nat Comm 2019.

Rather I share the authors’ perception that they might have captured the same signal in both cohorts, but in COPSAC 2010 by proxies of the three pathogens. Therefore, I consider the new Supplementary Fig. 6 very helpful; panels a and b might be presented in the main manuscript.

R2: We are grateful for this pertinent comment from the reviewer. As suggested, we have now removed the Kaplan Meier curve (prev. fig 4c) and instead added a forest plot of the crude vs mutually adjusted estimates. Furthermore, we have added the heatmap (prev. supplementary fig 6A) as a new figure 5. We have kept the prev. panel b in the supplemental figure since it’s largely redundant with the former, and therefore more suitable as a supplemental figure. We have now referred to these changes in the results,

“Detection of either of these two genera was not associated with persistent wheeze/asthma by age 7 years, nor were they differentially abundant at the genus level (Fig. 4c-e). Adjusting analyses for the same covariates mentioned above did not change the results, but mutually adjusting the presence/absence of Veillonella/Prevotella and the pathogen score resulted in attenuation of the estimate for Veillonella/Prevotella but not the pathogen score (Fig 4e).” and

“We found that Veillonella and Prevotella were included in an apparent cluster with Streptococcus and Haemophilus (Fig 5 and Supplemental Fig. 6).”

C3: I share the other reviewer’s concerns that the study design is not suitable for comparing sampling sources and detection methods. However, I acknowledge that both COPSAC studies were not set up for such comparisons. Therefore, I recommend emphasizing that (1) such comparisons have been covered by other more suited studies and (2) the currently presented analyses are secondary analyses of studies with different original aims. The latter, however, involve the relation of airway pathogens and asthma development, which is in the focus of the current manuscript.

R3: We agree and have now emphasized these aspects in the discussion section:

“Importantly, our study was not designed to compare different upper airway sampling locations or detection methods, which has been performed elsewhere²⁰⁻²², but rather to examine early life risk factors for disease, in particular asthma.”

C4: With respect to response R6, I suggest shifting the focus from the difference between the studies

towards the common finding of the pathogen cluster. In my opinion, consistency between studies can be demonstrated without explaining the differences between them. Besides, the studies are underpowered to exclude differences. Insofar, the statement “We can establish based on these results that the association between pathogenic bacteria and asthma in COPSAC2000 did not critically depend on sampling location and method nor detection method” is too strong.

R4: We have now modified the section and toned down the wording and shifted focus to include this consistency. The section now reads,

“We can establish based on these results that the association between pathogenic bacteria and asthma in COPSAC₂₀₀₀ recapitulated the culture-based findings independent of sampling location & method and detection method. While we did find overlaps between the taxa representing the pathogenic score and Veillonella and Prevotella, still some unresolved cohort differences exist.”

C5: The argument in lines 226-229 is not logical as it mixes up the issues of temporality and confounding. Besides I would not consider pharmacological treatment in a cross-sectional study more problematic than the intervention performed in COPSAC2010.

R5: The intent with this comment was comparing the longitudinal design (recruiting healthy newborns) with case-control studies of patients vs healthy controls, where there are numerous differences (incl treatment) as part of the study design. However, we agree that this was not formulated in a good way – of course the reviewer is correct that confounding is still an important challenge. Therefore, we have removed the sentence from the manuscript.

Minor:

C6: It might be easier for the reader to state in the figure legends which study is the basis of the analyses.

R6: We have now added this information to the fig 2+3 legends, and the new fig 5. The information was already included in figs 1+4.

C7: The use of “abundance” in the plural is jargon and might be replaced by “abundance values”.

R7: We agree and have changed this term throughout the manuscript.